

# Using a convection-permitting climate model to predict wine grape productivity: two case studies in Italy

Laura T. Massano[1], Giorgia Fosser[1], Marco Gaetani[1], Cécile Caillaud[2]

[1]Scuola Universitaria Superiore IUSS, Pavia, 2700, Italy

2CNRM Centre National de Recherches Météorologiques CNRM, Groupe de Météorologie de Grande Échelle et Climat

*Correspondence to*: Laura T. Massano (laura.massano@iusspavia.it)

**Abstract.** Viticulture is tied to climate, it influences the suitability of an area, yield and quality of wine grapes. Therefore, traditional wine-growing regions could be threatened by a changing climate. Italy is at-risk being part of the Mediterranean climatic hotspot and judged in 2022 the second-largest exporter of wine worldwide. The article

explores the potential of climate models to predict wine grape productivity at local scale. To this end, both single and multi-regression approaches are used to link productivity data provided by two Italian wine consortia with bioclimatic indices. Temperature and precipitation-based bioclimatic indices are computed by using the observational dataset E-OBS, the high-resolution climate reanalysis product SPHERA, and both the Regional and the Convection-permitting Climate Model (RCM and CPM). The potential of CPMs to represent the impact of climate variability on wine grape

productivity at local scale in Italy is evaluated. The results indicate high correlations between some bioclimatic indices and productivity. Climate models appear to be a useful tool to explain productivity variance, however, the added value of CPM, became evident only when precipitation-based indices are considered. This assessment opens the path for using climate models, especially at convection-permitting scale, to investigate future climate change impact on wine production.

## 1 Introduction

Wine-growing has a strong socio-economic impact and is one of the principal agricultural economic activities in Italy, that in 2022 was the world's leading wine producer (49.8 million hl), and second largest wine exporter, with a value of 7.8 billion euros.

Climate plays a significant role in viticulture, determining the suitability of an area and influencing wine grape yield

and quality. Over the coming decades, the wine sector is expected to be affected by climate change especially in Italy that is part of the Mediterranean climatic hotspot (Tuel and Eltahir, 2020), where the impact of climate change is expected to be more severe than the global average (Bernetti et al., 2012; Sacchelli et al., 2016). In this context, many studies investigated the impact of rising temperatures and changing rainfall patterns on grape growth (Bagagiolo et al., 2021; Gentilucci, 2020). Temperature is the primary driver for the phenological phases (Fraga et al., 2016), and a

warmer climate may lead to an earlier onset of phenological phases and to a shorter growing cycle, increase frost-related risks, as budburst occurring earlier in spring, when frost events are still frequent (Lamichhane, 2021; Trought et al., 1999). Furthermore, traditional wine-producing regions, as Douro in Portugal, La Rioja in Spain, Bordeaux in



France, and Tuscany in Italy, are expected to experience important shifts in viticulture suitability that can consequently causes a decline in production (Adão et al., 2023; Rafique et al., 2023; Sgubin et al., 2023; Tóth and Végvári, 2016).

A common tool to investigate the impact of climate variability and change on the wine sector is the use of bioclimatic indices, developed from climate variables for specific plants and crops (Badr et al., 2018; Chou et al., 2023; Gaitán and Pino-Otín, 2023). A set of bioclimatic indices, based on temperature and heat accumulation (OIV, 2015), was proposed by the International Organisation of Vine and Wine (OIV), while precipitation-based indices were developed by Badr et al., (2018) considering the research of Blanco-Ward et al. (2007). Bioclimatic indices are commonly used

to assess a region's suitability for viticulture or zoning purposes, as well as in relation to phenology, harvest date and alcohol concentration (Dalla Marta et al., 2010; Koufos et al., 2014; Sánchez et al., 2019; Teslić et al., 2018). A novel application linking bioclimatic indices directly to wine grape productivity data in Italy was proposed by Massano et al., (2023) at regional level.

In Italy the vineyards are planted in extremely different areas, from the coasts to the hills, in some case also at

considerable altitude (Tarolli et al., 2023). The wine production system is complex and fragmented, including both small farms and large companies. To valorise the designation of origin and guarantee a defined level of quality, producers are organized in wine consortia (Consorzi di Tutela) according to the EU and national regulations (e.i. Regulation (EU) No 1308/2013, Disciplinari regionali) (Gori and Alampi Sottini, 2014; Ugaglia et al., 2019). To address this fragmentation and account for the typicity of the wine business (Agnoli et al., 2023; Spielmann and

Charters, 2013), yield data from the wine consortia and high-resolution climate data are of prominent importance for local-scale impact studies and, thus for effective adaptation strategies.

In the context of impact studies at local scale, requiring high-resolution climatic data, the use of km-scale convection permitting models (CPM) is increasing (Bamba et al., 2023; Le Roy et al., 2021; Tradowsky et al., 2023). Thanks to their high spatial resolution (less than 4 km), CPMs can represent convection explicitly without the need for

parameterisation, thus reducing the associated model uncertainty (Fosser et al., 2024). Compared to coarser resolution regional climate models (RCMs), the CPMs represent more realistically hourly rainfall intensity, the diurnal cycle of precipitation and the extremes and are thus consider more reliable in terms of climate projections of precipitation (Brisson et al., 2016; Coppola et al., 2020; Fosser et al., 2020, 2015; Kendon et al., 2017; Pichelli et al., 2021; Ban et al., 2021). The advantages of CPMs versus RCMs has been also explored in the assessment of the impact of climate

change on agriculture and crop production (Agyeman et al., 2023; Berthou et al., 2019; Chapman et al., 2020, 2023). This study assesses the potential of a CPM to represent the impact of climate variability on wine grape productivity at the local scale, by relating temperature and precipitation-based bioclimatic indices to wine productivity data provided by two wine consortia in northern and central Italy. The CPM performance is validated against climate observations and a reanalysis product, as well as compared to the driving RCM simulation to investigate the added-value of the

higher resolution. Single and multiple regression approaches are used to determine the extent to which bioclimatic indices can explain changes in wine grape productivity at local scale. The multiple regression approach accounts for the potential interplay between the bioclimatic indices, potentially increasing the portion of total productivity variability explained by the individual indices, as found by Massano et al. (2023).



## 2 Data and Methods

### 2.1 Wine grape data

Wine grape yield data, as well as the hectares devoted to viticulture, are collected from two wine consortia in Italy: 'Consorzio per la tutela del Franciacorta' (FRA) and 'Consorzio Del Vino Nobile di Montepulciano' (MON). The first one lies in Franciacorta, a small (200 km2) wine-growing region in Lombardia (LOM), in northern Italy, mostly known for sparkling wine (Figure 1a). The area is characterised by a humid subtropical climate according to the Koppen classification (Costantini et al., 2013). The Iseo lake, located at the northern border of this region, is the sixth largest lake in Italy and tempers the typical heat of the plain in summer, while in winter protects the vineyards from the freezing air arriving from the north (Leoni et al., 2019). The consortium was born in 1990 thanks to the endeavour of local producers that felt the need to preserve the original production method of the Franciacorta wine. Today the consortium is composed by 200 winemakers and preserves three designations: Sebino IGT (Typical Geographical Indication), Franciacorta DOCG (Denomination of Controlled and Guaranteed Origin) and Curtefranca DOC (Denomination of Controlled Origin), known as "Terre di Franciacorta" before 2011 (https://franciacorta.wine/en/). This analysis focuses on the designations of Franciacorta DOCG and Curtefranca DOC from 1997 to 2019 (23 years), discarding Sebino IGT, for which data are only available for a limited period.

a)                                          b)

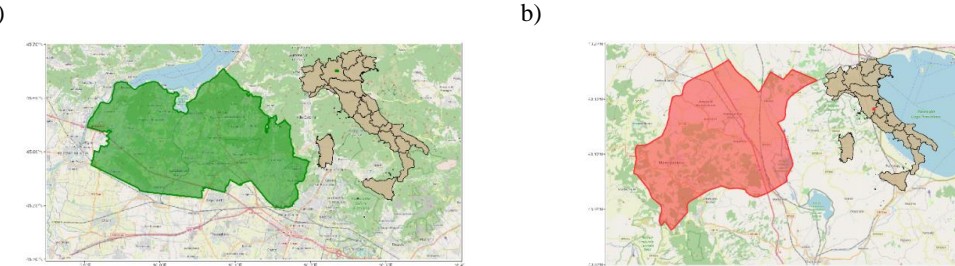

**Figure 1 a) Area of Franciacorta Consortium (FRA), Lombardia (LOM) region, North of Italy. b) Area of the Consorzio del Vino Nobile di Montepulciano (MON), Toscana (TOS) region, centre of Italy (base layer : © OpenStreetMap contributors 2019. Distributed under the Open Data Commons Open Database License (ODbL) v1.0.).**

The "Consorzio del Vino Nobile di Montepulciano" (MON) (https://www.consorziovinonobile.it/) is located within the Montepulciano territory in Toscana (TOS) region in the centre of Italy (Figure 1b). The area is characterized by a Mediterranean climate with hot and dry summer, and mild and rainy winters (Costantini et al., 2013). The consortium preserves three designations, namely Vino Nobile di Montepulciano DOCG, Rosso di Montepulciano DOC and Vin Santo di Montepulciano DOC. The study focuses on the first two designations that have the longest time series covering 31 years between 1989 and 2019.

For each wine designation, the FRA consortium directly reports the quantity of grapes harvested in quintals (q), while MON indicates the hectolitres of wine produced (hl) and the maximum percentage of the grape yield convertible into wine (70%). For the analysis, the hectolitres are converted into quintals using the maximum percentage allowed, and then the productivity (q/ha) is calculated by dividing the quintals of grapes by the vineyard area.

To check the consistency of productivity data between local and regional scales, and thus contextualise this work within the broader framework of previous studies (e.g. Di Paola et al., 2023), the productivity at the local scales (FRA



and MON) is compared with the productivity at regional scale provided by the Italian National Institute of Statistics

(ISTAT). ISTAT provides the harvested wine grape (in quintals) and the area devoted to vines (in hectares) from 1980 onwards. However, the data are not homogenous over time in terms of spatial aggregation. Wine grape productivity data are available at the provincial level between 1980 and 1993 and from 2006 to 2019; at regional level between 1994 and 2000; at national scale while from 2000 to 2005. Following Massano et al (2023), the data were aggregated at regional level for Lombardia (LOM) and Toscana (TOS) region, where the FRA and MON consortia are

respectively located, for the period 1980–2019, with a six-year gap between 2000 and 2005. Considering the overlapping periods between ISTAT and consortia time series, it is found that the regional and local productivity data are significantly correlated ($p<=0.05$) for both FRA and MON (Table A 1). In addition, the Welch's t-test proves that both consortium distributions are part of the regional population (Table A 1 and Figure A 1).

### 2.2 Observational climate data

The observational dataset used is E-OBS, a gridded daily data set covering Europe from January 1950 to the present day. E-OBS is constructed using data from meteorological stations provided by the European National Meteorological and Hydrological Services (NMHSs) or other data holding institutions (Photiadou et al., 2017; Van Der Schrier et al., 2013). The analysis is based on the latest available version (v28) at 0.1 deg (~11 km). Although the E-OBS database is frequently used to validate climate models (Lorenz and Jacob, 2010; Retalis et al., 2016; Christensen et al., 2008;

Jaeger and Seneviratne, 2011) , some studies have pointed out limitations in the E-OBS representation of precipitation and temperature, mainly due to the inhomogeneity of the station network used for interpolation (Kyselý and Plavcová, 2010; Van Der Schrier et al., 2013; Liakopoulou and Mavromatis, 2023).

In addition to observations, the analysis uses a high-resolution convection-permitting reanalysis product, called SPHERA (High rEsolution ReAnalysis over Italy; Cerenzia et al., 2022; Giordani et al., 2023), produced by ARPAE-

SIMC (Agency for Environmental Protection of the Emilia Romagna Region, Italy). Based on the non-hydrostatic limited-area model COSMO (Schättler et al., 2018; Baldauf et al., 2011), SPHERA dynamically downscales the global reanalysis ERA5 (Hersbach et al., 2020) assimilating regional in situ observations to improve the quality of the simulation. This new reanalysis product covers Italy at a horizontal resolution of 2.2 km with a temporal coverage of 26 years (1995-2020). SPHERA reanalysis, validated against ERA5 by Giordani et al. (2023), shows added value for

the description of moderate to severe local precipitation events and extreme rainfall. The performance of SPHERA demonstrates that it can be a valuable resource for improving climate monitoring by providing insights into regional climate change impacts (Giordani et al., 2023).

### 2.3 Climate model data

The French Centre National de Recherches Météorologiques (CNRM) provides two climate simulations for the period

2000-2018. The first simulation is based on an RCM model, CNRM-ALADIN (Nabat et al., 2020), covering the Med-CORDEX domain, driven by the ERA-Interim (80 km) reanalysis (Dee et al., 2011), while the second one is based on a CPM model, CNRM-AROME, covering the pan-Alpine domain defined within the CORDEX FPS on Convection programme (Lucas-Picher et al., 2023; Coppola et al., 2020).  CNRM-ALADIN (hereafter RCM) has a horizontal





resolution of 12.5 km and is the limited area version of ARPEGE-Climate. CNRM-AROME (hereafter CPM), is a
convection-permitting model dynamically downscaled form CNRM-ALADIN, with a resolution of 2.5 km. CPMs are
kilometer-scale regional climate models, with typically horizontal gridding of less than 4 km, which allows a more
accurate representation of surface and orographic features. They are also non-hydrostatic models that can explicitly
resolve deep convection and therefore better represent convective phenomena, such as heavy convective precipitation.
Further information on these climate model simulations can be found in (Caillaud et al., 2021)

**2.4 Bioclimatic indices**

This study considers ten bioclimatic indices (summarised in Table 1): eight of them, recommended by the International
Organisation of Vine and Wine (OIV), are based on temperature and heat accumulation, while the other two are based
on rainfall accumulation.

The temperature-based indicators are:

1. Daily mean temperature during vegetation period (TmVeg) calculated between 1st April to 31st October (Jones et
al., 2005). Temperature in spring plays a key role in determining the timing of the phenological events, as underlined
by Malheiro et al., (2013). In general, higher TmVeg leads to an anticipation of the phenological phases, while TmVeg
values above 24 °C or below 12 °C are considered unfavourable to wine-growing (Eccel et al., 2016).

2. Heliothermic Huglin index (HI), which is calculated by summing, when positive, the average between the mean
and the maximum temperature, in relation to the baseline temperature of 10°C i.e. the physiological threshold for the
start of the vine growth cycle (Huglin M, 1978; Teslić et al., 2018), over the period from 1st April to 30th September
and corrected by a coefficient of day duration. The HI index is tied to vine growing and grape sugar concentration
with higher HI leading to an increased vine vigour and higher sugar content in the grapes. According to Tonietto and
Carbonneau (2004), a climate with a heat index (HI) of more than 3000 degrees per day is classified as 'very warm',
while below 1200 degrees per day is "too cold". Both these situations are associated to plant stress and thus lead to a
production reduction.

3. Winkler degree days (WI), which provides a measure of heat accumulation during the growing season, is the sum
of daily mean temperatures above 10°C from 1st April to 31st October (Amerine and Winkler, 1944; Piña-Rey et al.,
2020). Similarly, to HI, WI index is linked to the rate of growth of the vines and the development of the fruits, with
values between 850 and 2700 degree days being optimal for the wine production (Eccel et al., 2016).

4. Biologically Effective Degree Days (BEDD), which is the sum of daily mean temperatures in the range between 10
°C and 19 °C, from 1st April to 31st of October. The BEDD index uses the same baseline temperature (10 °C) as WI
and HI indices but also take into consideration that vine growth is unlikely to occur above the upper temperature
threshold of 19°C (Anderson et al., 2012; Gladstones, 1992). As the previous temperature-based indices, too high
(above 2000 degrees per day) or too low (below 1000 degrees per day) values of BEDD can potentially reduce
productivity.

5. Cool Night Index (CNI), defined as the average minimum air temperature during the month of September. Low
minimum temperatures in September increase the polyphenolics in the grapes and are beneficial for the overall quality



of the harvest (Tonietto and Carbonneau, 2004). Although CIN is more related to grape quality than quantity, Massano
et al (2023) found that this index can help explaining changes in productivity especially when used in combination
with other bioclimatic indices.

6. Minimum temperature during vegetative period (TnVeg), which is the minimum temperature recorded during the
vegetative period (1st April to 31st October). This index is important to assess the occurrence of spring frosts that
pose a significant risk to viticultural practices and production. The damage threshold is fixed at -2 °C (Sgubin et al.,
2018).

7. Maximum temperature during vegetative period (TxVeg), which is the maximum temperature recorded during the
vegetative period. This index is useful for assessing the occurrence and the severity of summer hot-spells that can
damage to vineyard, thus reducing the wine productivity (Cabré and Nuñez, 2020). The heat stress threshold is set at
35°C, above which physiological damage to the vines is expected (Hunter and Bonnardot, 2011).

8. Minimum temperature during rest period (TnRest), defined as the minimum temperature during rest period, i.e. 1st
November to 31st March. This index is used to determine winter severity. Grapevines can tolerate temperatures as -
25 °C (Düring, 1997; Lisek, 2012), although damage can already occurs at -15 °C  (Eccel et al., 2016)

The indices based on precipitation are:

1. Growing season precipitation index (GSP), defined as rainfall accumulated from 1st April to 30th September and
used to assess the water stress for non-irrigated grapevines (Blanco-Ward et al., 2007; Piña-Rey et al., 2020), as in
Italy where irrigation is only allowed in extreme cases (e.g. long drought periods).

2. Spring Rain index (SprR), which measures the amount of rain accumulated between the 21st of April and the 21st
of June (Raül Marcos-Matamoros et al., 2020). This indicator of spring wetness can be related to production. In fact,
while dry springs can delay vegetative growth, wet ones can increase plant vigour but also lead to a higher risk of
fungal diseases (Alessandro Dell'Aquila, 2022).

**Table 1: Acronyms and formulas of the bioclimatic indices used.**

| | Definition | Formula | Suitable class range |
|---|---|---|---|
| Temperature-based | Mean temperature during vegetation period (TmVeg) | $TmVeg = T_{mean}$ (1) <br><br> *between 1st April and 31th October* | 13-24 °C <br><br> (Eccel et al., 2016) |
| | Heliothermic Huglin index (HI) | $HI = K \sum_{01Apr}^{30Sep} max\left[\left(\frac{(T_{mean}-10)+(T_{max}-10)}{2}\right); 0\right]$ (2) <br><br> K=1.04 length of days coefficient | 1200-3000 °C <br><br> (Tonietto and Carbonneau, 2004) |
| | Winkler degree days (WI) | $WI = \sum_{01Apr}^{31Oct} max\left[\left(\frac{T_{min}+T_{max}}{2} - 10\right); 0\right]$ (3) | 850-2700 °C <br><br> (Eccel et al., 2016) |
| | Biologically Effective Degree Days (BEDD) | $BEDD = \sum_{01Apr}^{31Oct} min\{max\left[\left(\frac{T_{min}+T_{max}}{2} - 10\right); 0\right]; 9\}$ (4) | 1000-2000 °C <br> (Gladstones, 1992) |
| | Cool Night Index (CNI) | $CNI = \frac{1}{30}\sum_{01Sep}^{30Sep} T_{min}$ (5) | 12-18 °C (Tonietto and Carbonneau, |





| | | | 2004) |
|---|---|---|---|
| | Minimum temperature during vegetative period (TnVeg) | $TnVeg = T_{min}$ between 01 Apr and 31 Oct (6) | Damage threshold - 2 °C (Sgubin et al., 2018) |
| | Maximum temperature during vegetative period (TxVeg) | $TxVeg = T_{max}$ between 01 Apr and 31 Oct (7) | Upper threshold 35 °C (Hunter and Bonnardot, 2011) |
| | Minimum temperature during rest period (TnRest) | $TnRest = T_{min}$ between 01 Nov and 31 Mar (8) | Above -25 °C (Düring, 1997; Lisek, 2012) |
| Precipitation-based | Growing season precipitation index (GSP) | $GSP = \sum_{01Apr}^{30Sep} Prec$ (9) <br><br> *Prec: total precipitation* | 200-600 mm <br><br> (Badr et al., 2018) |
| | Spring Rain index (SprR) | $SprR = \sum_{21Apr}^{21Jun} Prec$ (10) | (Dell'Aquila, 2022) |

**2.5 Validation of climate simulations and calculation of bioclimatic indices**

In this work, temperature and precipitation data from the observational dataset E-OBS, the climate reanalysis product SPHERA and the climate model simulations, at regional (RCM) and convection-permitting scale (CPM), are used for the calculation of the above-described bioclimatic indices. The analysis focuses on the 19 years from 2000 to 2018 that is the longest period available for RCM and CPM simulations and in common with E-OBS, SPHERA as well as FRA and MON productivity data.

To compare the observational datasets and climate simulations among each other (Berg et al., 2013), they are first all remapped on a common grid, i.e. E-OBS regular grid, at ~11 km. Tests performed to investigate the effects of the remapping strategy on the climate variables showed that the results are not impacted by the resolution chosen for the remapping (not shown).

Then, the climatic variables (i.e. P: Precipitation; TM: mean temperature, TX: max temperature and TN: min temperature) are retained on all available grid cells within the areas of interest (LOM and TOS). Subsequently, the consortium territory is cropped using the respective shape files of FRA and MON. Finally, the spatial average is calculated by weighing the contribution of each grid cell according to the percentage of the cell falling within the consortium territory. The shape file of the FRA consortium's territory is provided directly by the consortium's technical office, while the shape file for MON is created by selecting the municipality listed in the appellation regulation for the relevant denominations (i.e., Montepulciano municipality). The same methodology is used to calculate the bioclimatic indices.

The precipitation and temperature time series of the climate simulations are analysed against the observational datasets to evaluate the biases in the climatic conditions in the region of interest, prior to examine the bioclimatic indices. In



particular, the CPM performance is evaluated for the common period 2000-2018 against both SPHERA and E-OBS and compared to the RCM. In this study, the new SPHERA reanalysis product is used as a reference dataset together with the E-OBS dataset, which is already widely used for model validation (Kyselý and Plavcová, 2010). SPHERA and E-OBS time series together provide a range within which the CPM and the RCM time series are expected to fall, similar to a 'confidence interval'.

The comparison between SPHERA (E-OBS) and CPM, as well as SPHERA (E-OBS) and RCM, is carried out by computing the Spearman correlation and RSME, the percentage differences of RMSE with the mean of the reference (SPHERA and E-OBS) (RMSE%) is also indicated for the cumulable variables (i.e. BEDD, HI, WI,GSP, SprR and precipitation). This allows to analyse whether the variability of SPHERA and E-OBS data is reproduced by CPM and RCM simulations and asses the biases between model simulations and both reanalysis and observations. Statistical significance of the differences between model simulations and both reanalysis and observations is assessed by a Welch's two-tailed t-test, with a 95% level of confidence.

Finally, a trend analysis for both the climatic variables and the bioclimatic indices is performed to assess the evolution of the climatic condition in FRA and MON in the period 2000-2018; the same analysis is also carried out for productivity data. The non-parametric Mann-Kendall test and the Sen's slope estimator are used to determine the presence and the magnitude of trends with a significance level of 95% (Hanif et al., 2022; Mann, 1945). The assessment of possible trends aims to investigate whether the long-term component of variability may be dominant over the interannual component.

## 2.6 Single and multi-regression approach

The Spearman correlation coefficient between each bioclimatic index and wine grape productivity is calculated for both consortia area and the threshold for statistical significance is set to 95%. This analysis aims at assessing the fraction of wine grape productivity variability explained by the bioclimatic indices and the ability of climate models to represent this relationship compared to the observational datasets.

Furthermore, a multi-regressive (MR) approach is applied to determine whether a linear combination of indices can enhance the total productivity variability explained by the bioclimatic indices (Massano et al., 2023). The best subsets regression technique is used to establish the most suitable combination of indices. This approach seeks the predictor subset of bioclimatic indices that most accurately predicts the outcome variable, i.e. the productivity. It examines all feasible predictor combinations and removes irrelevant ones to streamline the model. The k-fold cross validation method is employed to identify the optimal model (Kassambara, 2017). This method performs cross-validation by randomly dividing the data into k subsets (k-fold) approximately of equal size, with k typically set to 5 or 10 (here k = 5 is used). One of the folds serves as test set and the remaining as training. This process is repeated k times, whereby varying groups of data are utilized as training or testing sets. Subsequently, the mean squared error is computed. The average of the mean squared errors of all iterations is the model prediction error (CV - cross validation error) (James et al., 2021; Kuhn and Johnson, 2013; Wassennan, 2004). The performance of the multi regressive model is assessed



by the adjusted R-squared coefficient of determination (AdjR$^2$), while the p-value is used to determine statistical significance at 95% level. The so optimised multi-regression model is then used to predict the past productivity, which is compared to the observed productivity using the Pearson correlation. When the MR method provides statistically significant results, the variance explained by the MR model is compared with the maximum variance explained by SR to determine which method provides the best performances.

## 3 Results

### 3.1 Validation of the climate simulations

The precipitation and temperature time series of both CPM and RCM are validated against the observational datasets to evaluate the biases in the climatic conditions of the consortia (FRA and MON), which could lead to biases in the bioclimatic indices. To this end, Figure A 2 for FRA, and Figure A 3 for MON, show the precipitation (P) and temperature (TM: mean temperature, TX: max temperature and TN: min temperature) time series of E-OBS, SPHERA, RCM and CPM for the common period 2000-2018. In MON, E-OBS minimum temperature time series shows a strong decrease of almost 2°C between 2015 and 2018 (Figure A 3), which is not observed in any of the other datasets. Further investigations highlighted that this temperature fall affects the entire TOS and is inconsistent with other observational records (not shown). This E-OBS misrepresentation of the temperature field affects consequentially the mean temperature time series (Figure A 3), the temporal correlations (Table A 2), and is likely to be reflected in the temperature-based indices. Previous studies have shown that E-OBS underestimates monthly and seasonal average temperatures when compared to stations observations (Liakopoulou and Mavromatis, 2023). In general, both RCM and CPM show high and significant temporal correlations with SPHERA for all the climate variables in both consortia (Table A 2), indicating that RCM and CPM reproduce the same variability of SPHERA, although the climate simulations tend to overestimate mean and maximum temperature while underestimating the minimum, as reflected by the statistical differences in mean values (Table A 3). In FRA the variability observed in E-OBS is always reproduced also in RCM and CPM simulations. The Welch's t-test confirmed that E-OBS is closer in mean value to RCM than CPM simulations. Figure 2 and Figure 3 show the ten bioclimatic indices time series computed in the two consortia areas for E-OBS, SPHERA, RCM and CPM. All the bioclimatic indices show very high and significant temporal correlation between SPHERA and both RCM and CPM in both consortia (Table 2). Similar conclusion can be draw for the comparison of the climate models with E-OBS in FRA, while in MON four temperature-base indices (i.e. BEDD, WI, TnVeg, CNI) are not significantly correlated, likely due to the low correlations in medium and minimum temperature (Table A 2). The correlations, especially with SPHERA, tend to be slightly higher for the CPM than for the RCM for most indices, despite the higher RMSE in the CPM (Table 2). The strong correlation between SPHERA and climate simulations ( Table 2) indicates that RCM and CPM reproduce the same variability of SPHERA, despite the statistical differences in mean values (Table A 4). The same conclusion is valid also for the comparison of RCM and CPM to E-OBS. This analysis suggests both CPM and RCM could be a valid alternative to observational datasets to investigate the impact of climate on viticulture, despite the biases affecting the climate simulations.





**Figure 2: Bioclimatic indices time series 2000-2018, averaged on the FRA consortium area.**





**Figure 3 Bioclimatic indices time series 2000-2018, averaged on the MON consortium area.**





**285** **Table 2: Spearman correlation coefficient the root mean square error (RMSE) of the indices time series and the percentage differences of RMSE with the mean of the reference (SPHERA and E-OBS) (RMSE%) for the cumulative variables. Bold font and asterisk (*) indicate a statistically significant result (p>=0.05)**

| FRA | | | | | | | | | | | | |
|---|---|---|---|---|---|---|---|---|---|---|---|---|
| | SPHERA vs CPM | | | SPHERA vs RCM | | | E-OBS vs CPM | | | E-OBS vs RCM | | |
| Index | ρ | RMSE | RMSE% | ρ | RMSE | RMSE% | ρ | RMSE | RMSE% | ρ | RMSE | RMSE% | Index |
| BEDD (GDD) | **0.97*** | 26.62 | 1.8 | **0.96*** | 19.39 | 1.3 | **0.85*** | 37.29 | 2.5 | **0.91*** | 45.78 | 3 | BEDD (GDD) |
| HI (GDD) | **0.98*** | 305.88 | 13.7 | **0.96*** | 308.59 | 13.8 | **0.88*** | 128.56 | 5.2 | **0.87*** | 117.36 | 4.7 | HI (GDD) |
| WI (GDD) | **0.99*** | 264.91 | 14 | **0.98*** | 247.63 | 13.1 | **0.85*** | 209.55 | 10.7 | **0.85*** | 191.23 | 9.7 | WI (GDD) |
| TmVeg (°C) | **0.99*** | 1.24 | - | **0.98*** | 1.14 | - | **0.85*** | 0.98 | - | **0.84*** | 0.87 | - | TmVeg (°C) |
| TnVeg (°C) | **0.63*** | 1.4 | - | **0.95*** | 2.59 | - | **0.65*** | 1 | - | **0.72*** | 1.53 | - | TnVeg (°C) |
| TxVeg (°C) | **0.81*** | 5.11 | - | **0.48*** | 4.42 | - | **0.52*** | 3.56 | - | **0.64*** | 2.77 | - | TxVeg (°C) |
| CNI (°C) | **0.95*** | 0.81 | - | **0.87*** | 1.24 | - | **0.85*** | 1.2 | - | **0.85*** | 0.91 | - | CNI (°C) |
| TnRest | **0.81*** | 0.76 | - | **0.85*** | 1.99 | - | **0.75*** | 2.14 | - | **0.8*** | 1.17 | - | TnRest |
| GSP (mm) | **0.64*** | 295.39 | 37.6 | **0.74*** | 410.3 | 52.3 | **0.5*** | 204.67 | 59.9 | **0.55*** | 103.91 | 30.4 | GSP (mm) |
| SprR (mm) | **0.91*** | 43.28 | 18.6 | **0.77*** | 65.38 | 28 | **0.68*** | 111.33 | 79.2 | **0.84*** | 57.54 | 40.9 | SprR (mm) |
| MON | | | | | | | | | | | | |
| | SPHERA vs CPM | | | SPHERA vs RCM | | | E-OBS vs CPM | | | E-OBS vs RCM | | |
| Index | ρ | RMSE | RMSE% | ρ | RMSE (°C) | RMSE% | ρ | RMSE | RMSE% | ρ | RMSE | RMSE% | Index |
| BEDD (GDD) | **0.92*** | 55.33 | 4 | **0.91*** | 51.04 | 3.7 | 0.35 | 96.32 | 6.4 | 0.43 | 96.27 | 6.4 | BEDD (GDD) |
| HI (GDD) | **0.86*** | 232.29 | 9.9 | **0.94*** | 233.54 | 10 | **0.82*** | 151.35 | 6.3 | **0.72*** | 158.76 | 6.6 | HI (GDD) |
| WI (GDD) | **0.93*** | 284.54 | 16.1 | **0.93*** | 284.39 | 16 | **0.45*** | 217.68 | 11.2 | 0.31 | 224.69 | 11.6 | WI (GDD) |
| TmVeg (°C) | **0.93*** | 1.34 | - | **0.92*** | 1.34 | - | 0.42 | 1.02 | - | 0.31 | 1.05 | - | TmVeg (°C) |
| TnVeg (°C) | **0.69*** | 0.94 | - | **0.77*** | 1.76 | - | **0.67*** | 1.36 | - | **0.62*** | 1.58 | - | TnVeg (°C) |
| TxVeg (°C) | **0.75*** | 2.75 | - | **0.83*** | 2.52 | - | **0.86*** | 2.02 | - | **0.82*** | 1.84 | - | TxVeg (°C) |
| CNI (°C) | **0.97*** | 0.84 | - | **0.95*** | 0.58 | - | **0.49*** | 1.9 | - | 0.4 | 1.38 | - | CNI (°C) |
| TnRest | **0.9*** | 1.43 | - | **0.86*** | 1.09 | - | **0.8*** | 1.94 | - | **0.79*** | 1.58 | - | TnRest |
| GSP (mm) | **0.48*** | 128.26 | 39.1 | **0.49*** | 106.85 | 32.6 | **0.71*** | 136.38 | 48.3 | **0.71*** | 45.89 | 16.2 | GSP (mm) |
| SprR (mm) | **0.84*** | 60.96 | 49.9 | **0.82*** | 40.48 | 33.1 | **0.75*** | 68.15 | 60.7 | **0.81*** | 34.61 | 30.8 | SprR (mm) |

## 3.3 Bioclimatic indices control on wine grape productivity

**290** ### 3.3.1 Single regression analysis

A Spearman correlation analysis is performed to investigate the relation between the different bioclimatic indices and wine grape productivity and consequently determine the amount of total productivity variability (interannual and long-term) explained by these indices.





In FRA, the correlation coefficients are similar between climate simulations, observations, and reanalysis for the
temperature-based indices, while diverge and are not significant for the precipitation-based ones (Figure 4). Few cases
are statistically significant: CNI with model simulations, SPHERA, and E-OBS; the BEDD index only when RCM
and E-OBS are used. In these cases, the long-term component of the total variability may be dominant, since BEDD,
CNI, as well as the FRA productivity, have significant trends (Table A 5). RCM presents a statistically significant
outcome also for TnRest, which does not show trend over the period 2000-2018. In this case, the interannual variability
might be more relevant to explain productivity. The statistically significant coefficients are all positive indicating a
positive effect on productivity of BEDD, CNI and TnRest.

In a previous study, conducted at regional scale using ISTAT productivity data and E-OBS (v26), Massano et al.
(2023) did not find any statistically significant correlations for LOM neither with temperature-based nor precipitation-
based indices. This indicates that working at a local scale is crucial to improve the portion of productivity variance
explained by the bioclimatic indices, while the use of CPM for FRA does not provide any advantage compared to the
RCM. Productivity data show a significant positive trend in FRA (Table A 6)

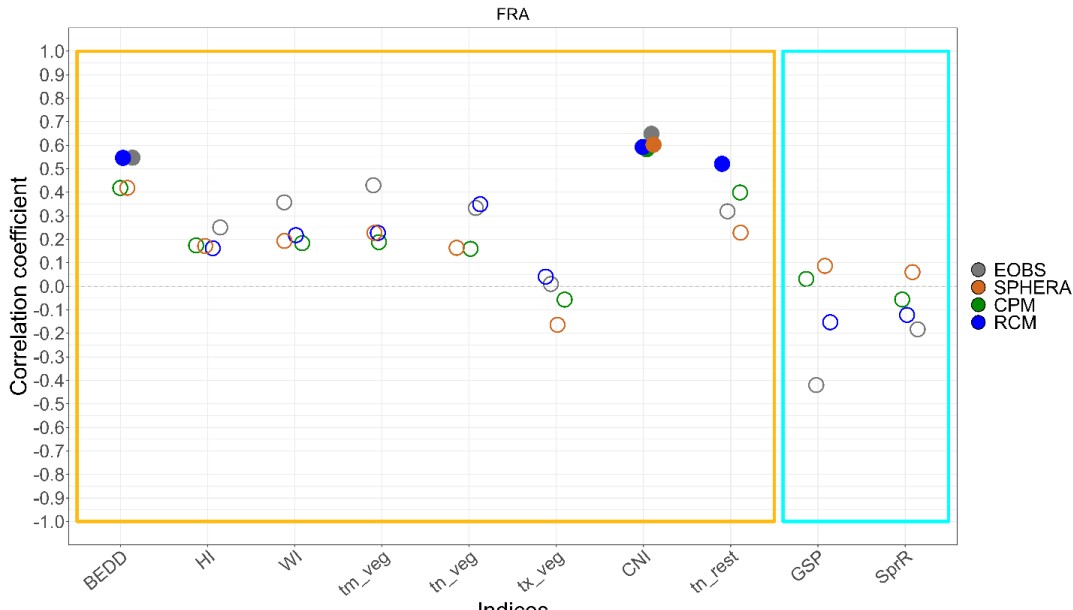

**Figure 4: Spearman correlations coefficients between bioclimatic indices and wine grape productivity in FRA. Full coloured
circles indicate significant correlations (p<=0.05).**

In MON, the correlations between productivity and bioclimatic indices are similar across all the datasets for BEDD,
HI, WI and TmVeg but show greater variation for all other temperature-based and the precipitation-based indices
(Figure 5). Significant results are found for TnVeg, only using CPM, and for TxVeg in all datasets. We highlight that
TxVeg displays a negative correlation, indicating that extreme temperatures during the growing period have a negative
impact on production. This aligns with wine makers expectations and is partially supported by results from FRA
(Figure 4), despite not being statistically significant. Both TnVeg and TxVeg indices show a significant positive trend



(Table A 7), which suggests productivity being more sensitive to the long-term component of variability, at least for CPM. Productivity data do not show any trend in MON (Table A 8).

Only the CPM simulation shows significant correlation for the precipitation-based index GSP. This could be linked to the more realistic representation of the precipitation field (Prein et al., 2015), although positive correlations with

GSP are not expected, as an excessively wet season is usually detrimental to production. Thus, it is possible that other factors influence this correlation, such as specific viticultural practices or vintage management (Priori et al., 2019). For example, harvesting immediately after rainfall may result in the collection of larger grapes, thus increasing the productivity. Additionally, specific trimming techniques can improve ventilation between the branches, reducing the risk of mould and fungus, and thus limiting the negative impact of precipitation on the harvest (Evers et al., 2010).

The MON case shows improvements compared to the analysis done with ISTAT data by Massano et al. (2023). In their analysis, TOS did not show any correlation between wine grape productivity and any bioclimatic indices, despite considering a longer time series. Being FRA and MON productivity data from the same population as the ISTAT productivity data (Table A 1 and Figure A 1), these results confirm that the use of the local scale and including a larger variety of bioclimatic indices can enhance the portion of productivity variability explained by the bioclimatic indices

considered.

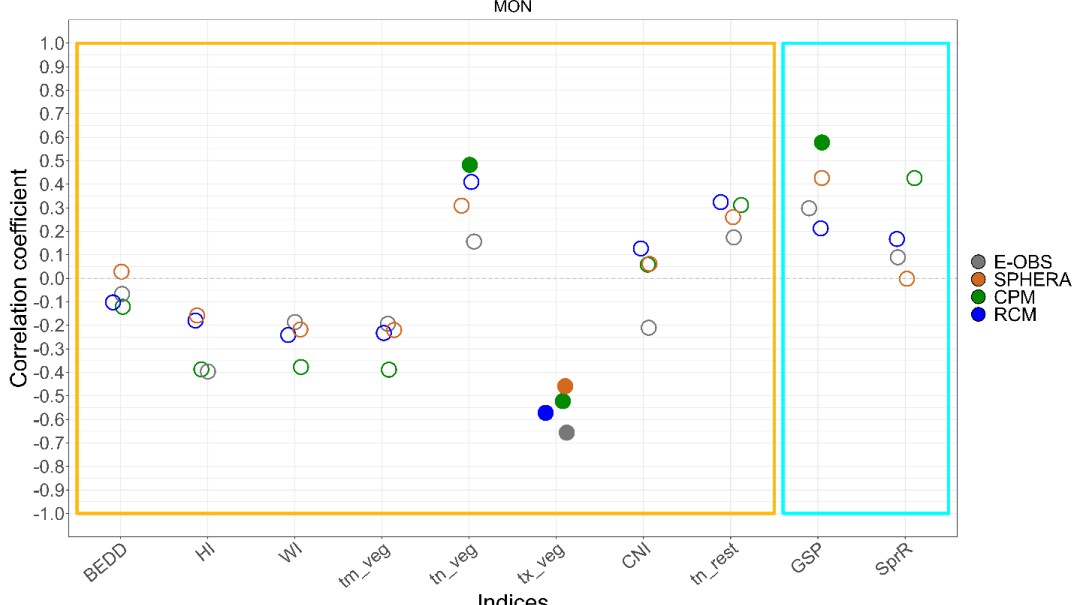

**Figure 5: Spearman correlations between bioclimatic indices and wine grape productivity in MON. Full coloured circles indicate significant correlations (p<=0.05).**




### 3.3.2 Multi-regression analysis

A multi regression (MR) analysis is carried out and compared with the single regression (SR) approach to see if considering a linear combination of bioclimatic indices increases the proportion of productivity variability explained by the indices.

Table 3 shows the results of the MR model, highlighting the selected bioclimatic indices and the variance explained in comparison with the SR method, for each case in both FRA and MON. The authors highlight that, even when the

340 MR selects just one index, this can differ from the single regression due to the correlation method chosen. The MR confirms that the temperature-based bioclimatic indices are more relevant than precipitation-based ones in explaining productivity variability, especially in FRA, where only for RCM the GSP is selected as a predictor. In MON, precipitation-based indices are selected as predictors in the MR model when using the CPM simulation and SPHERA reanalysis, confirming the relative higher importance of precipitation on productivity in this area compared to FRA.

Thus, for MON, the improved representation of the precipitation field at convection-permitting scale could be a relevant factor, since in the other cases precipitation-based indices are excluded by the MR.

**Table 3: Donuts chart indicating, for E-OBS, SPHERA, CPM and RCM, the best-performing index for the single regression (SR) and the indices included in the multi-regression model (MR), as well as the percentage of variance explained by each**
350 **model (centre of the donut), in FRA and MON. Orange (blue) colour indicates temperature-based (precipitation-based) indices. The MR Adjusted R2 is expressed in the MR Adj R$^2$ column.**

| | FRA | | | MON | | |
|---|---|---|---|---|---|---|
| **Data** | SR | MR | MR AdjR$^2$ | Data | SR | MR | MR AdjR$^2$ |
| **E-OBS** | var = 42 % | var = 35 % | 0.31 | **E-OBS** | var = 44 % | var = 32 % | 0.28 |
| **SPHERA** | var = 36 % | var = 56 % | 0.43 | **SPHERA** | var = 21 % | var = 42 % | 0.31 |
| **CPM** | var = 34 % | var = 48 % | 0.42 | **CPM** | var = 34 % | var = 45 % | 0.34 |



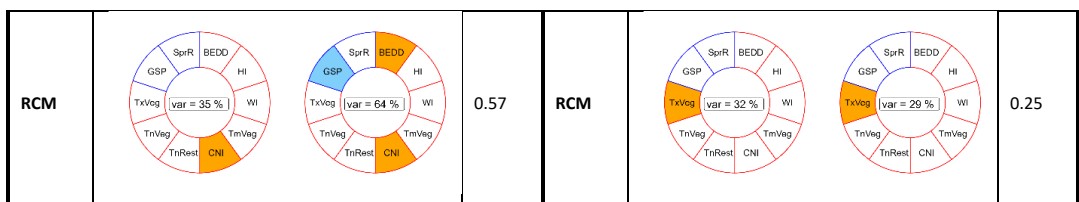

The overview on the performance of the single-regression method (SR) and the multi-regression method (MR) is presented in Figure 6, showing that using a linear combination of bioclimatic indices increases the proportion of explained total productivity variability, especially for FRA.

Overall, the bioclimatic indices explain a higher proportion of productivity variance in FRA compared to MON (Figure 6a and Table A 9), in line with previous findings at regional level for LOM and TOS (Massano et al., 2023). The highest proportion of explained variance in productivity is obtained in FRA with the MR approach and RCM data (64%), followed by SPHERA (56%) and CPM (48%). The variance explained in MON is lower, with a maximum of 45% obtained for CPM and the MR approach, very close to SPHERA with MR (42%) and to E-OBS with SR (44%). The maximum variance in productivity explained by the SR is compared with the MR variance (Figure 6b), demonstrating that the MR better represents productivity variability in FRA in all cases except E-OBS, which shows a slight decrease in performance (-7%). Meanwhile, SPHERA gains 20%, CPM 14% and RCM 29% when MR is compared to SR. In MON, MR provides a better explanation for productivity variance in SPHERA reanalysis and CPM simulation, accounting for an increase of 11% and 21% respectively. However, for the E-OBS dataset and RCM simulation, MR performance decreases slightly (-12% and -3% respectively).

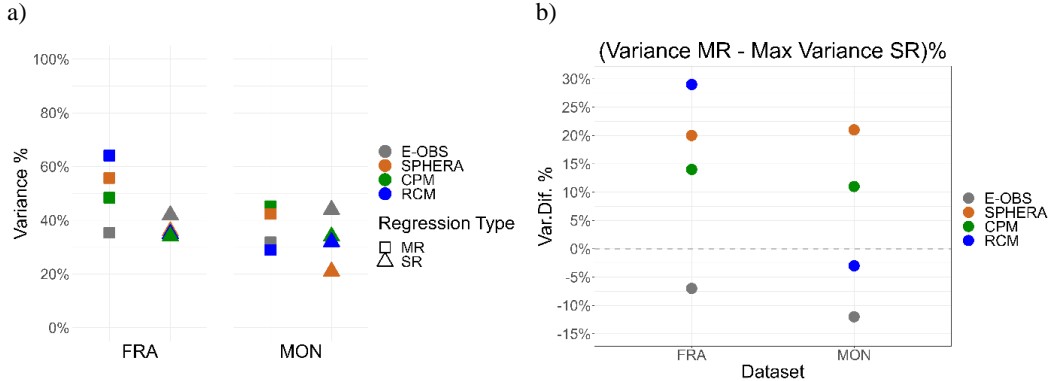

**Figure 6: a) The maximum fraction of the wine grape productivity variance (%) explained by SR and MR in each consortium, colours indicate the type of climatic data used, squared (triangular) shape indicates multi regressive (single regressive) approach. b) Variance differences in percentage between MR and SR for FRA and MON.**

## 4 Discussion and conclusion

The study assesses the potential of a CPM to investigate the impact of climate variability on wine grape productivity at a local scale, using bioclimatic indices for the period 2000-2018. The CPM simulation is compared with RCM





simulation, SPHERA reanalysis, and E-OBS observations. The study uses wine grape productivity data from two
Italian wine consortia, namely 'Consorzio per la tutela del Franciacorta' (FRA) and 'Consorzio Del Vino Nobile di
Montepulciano' (MON). Single and multiple regression approaches are used to account for the possible interplay of
bioclimatic indices in explaining wine grape productivity variability.

Overall, the single regression exhibits high correlation coefficients, but statistically significant results are only found
in a small number of cases at the 95% confidence level. The multi-regression method consistently enhances the
productivity variability explained by the bioclimatic indices and delivers predictors with potential for usability.

In FRA, the correlation coefficients are exclusively positive, but statistically significant only for temperature-based
indices such as BEDD, CNI, and TnRest. There is a high degree of agreement between CPM and SPHERA reanalysis,
but E-OBS and RCM presents the highest correlation. Correlations with precipitation-based indices in FRA are not
significant and tend to show negative relationships with productivity. These findings suggest that temperature is the
main factor affecting production, while precipitation has a negative impact on productivity, potentially resulting in
losses due to fungal diseases in the region.

The MON results indicate that only CPM provides statistically significant results for a precipitation-based index
(GSP), which is positively correlated with productivity. Also, SPHERA, RCM and E-OBS in this region show positive
correlations between precipitation-based indices and productivity, even if they are not significant. This differs from
the findings for FRA, where the correlations are negative, even if not significant. However, it is worth noting that
there are many differences in the geographical features and types of wine produced in FRA and MON. FRA is in the
humid subtropical climatic zone, while MON is situated in the hot summer Mediterranean zone. Other factors, such
as vintage management techniques and cultivar selection, can also influence productivity variability in addition to
climate, but investigation of these factors is beyond the scope of this paper. Meanwhile, the productivity for both FRA
and MON exhibits a negative correlation with TxVeg with all the climatic data considered, but it is only significant
for MON. This suggests that extreme maximum temperatures during the vegetative season (1$^{st}$ April - 30$^{th}$ October)
may have harmful effects.

These results, which are obtained at a local scale using data from consortia, complement the previous study conducted
at regional scale by Massano et al. (2023). The climate models appear to be a useful tool to explain productivity
variance using a MR approach, improving the results compared to the E-OBS. However, the use of the CPM does not
show a clear added value compared to the RCM since it performs better in MON, but not in FRA. This could be link
to the fact that temperature is generally the main driver of wine grape production, and the added value of the CPM
may be more evident when precipitation is a dominant factor.

However, the analysis presented here pave the path to the use of climate models to investigate the impact of climate
change on wine production in the future. In this context, CPMs can provide new climate information, such as hail risk,
which is a convections-related phenomenon that impact grape productivity. Moreover, this work shows an application
of the bioclimatic indices to wine grape productivity that is rarely used.

**Data availability**

Data can be provided by the corresponding authors upon request.



**Author contribution**

LM, MG and GF Conceptualization and Methodology, LM Formal analysis and Writing – original draft preparation, GF MG and CC Writing – review & editing.


**Competing interests**

The authors declare that they have no conflict of interest.

**Acknowledgments**

The work presented in this paper has been developed within the framework of the project "Dipartimento di Eccellenza
2023-2027", funded by the Italian Ministry of Education, University and Research at IUSS Pavia.

The authors gratefully acknowledge the WCRP-CORDEX-FPS on Convective phenomena at high resolution over Europe and the Mediterranean [FPSCONV-ALP-3]. This work is part of the Med-CORDEX initiative (http://www.medcordex.eu).

The authors want to express sincere gratitude to the 'Consorzio per la tutela del Franciacorta' and the 'Consorzio Del
Vino Nobile di Montepulciano' for the invaluable contribution provided in supplying the necessary data for this study.

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





**Appendix A**

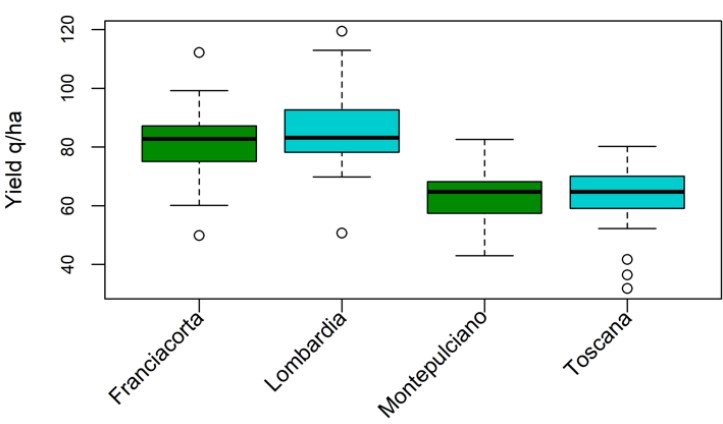


**Figure A 1: Boxplot of the regional productivity (cyan) and consortia productivity (green). The series of LOM and TOS come from ISTAT database and cover the period 1980-2019, whit a six-year gap between 2000-2005, the period available for FRA is 1997-2019 (calculated by aggregating the Franciacorta DOCG and Curtefranca DOC denominations) and for MON is 1989-2019 (calculated by aggregating the Vino Nobile and Rosso di Montepulciano denominations), with no gap in**
**the series.**

**Table A 1: results of Welch's t test (t.stat), the reference value for t.stat (t.tab), the degrees of freedom (DoF) for the t test based on the number of observations computed according to the Welch's equation for effective degrees of freedom (Welch, 1947) and temporal correlation between regional ad consortia productivity data. The * symbol indicates statistically significant results (p<=0.05).**

| Consortium | *t.stat* | *t.tab* | *DoF* | *Cor.Coef.* |
|---|---|---|---|---|
| FRA | 1.17 | 2.01 | 47.94 | 0.62* |
| MON | 0.1 | 2 | 63.99 | 0.55* |




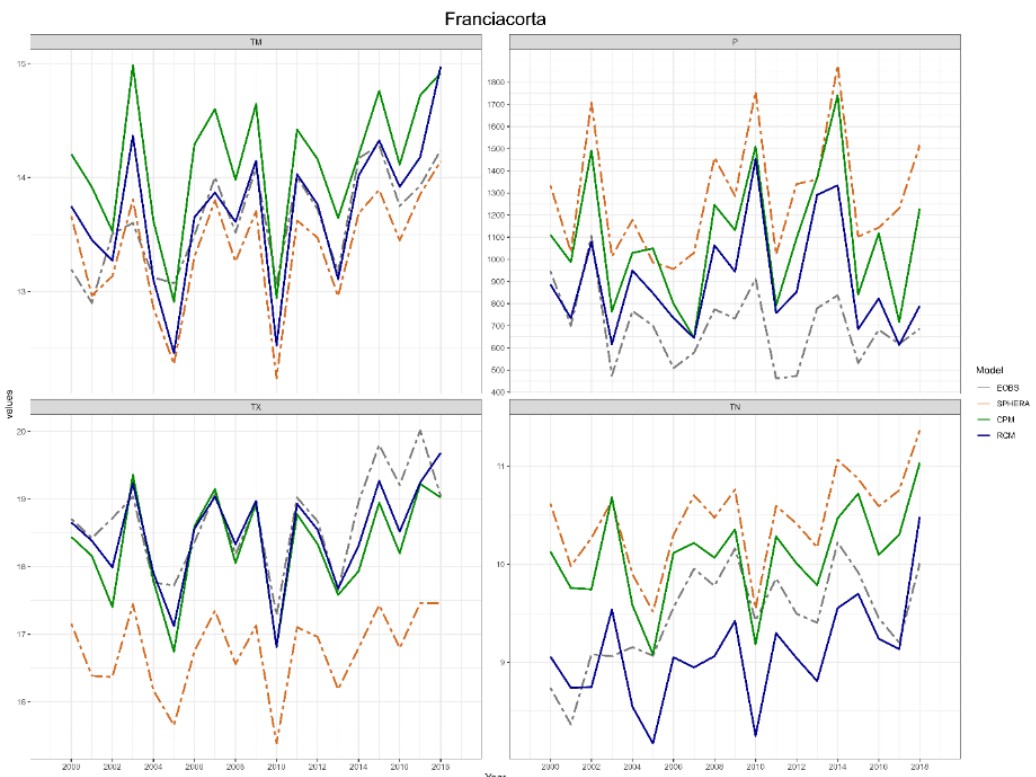

**Figure A 2: Time series of mean (TM), maximum Temperature (TX), minimum (TN) temperature and precipitation (P) over FRA area for the period 2000-2018. All the time series are based on data remapped on E-OBS grid (~ 11 km).**



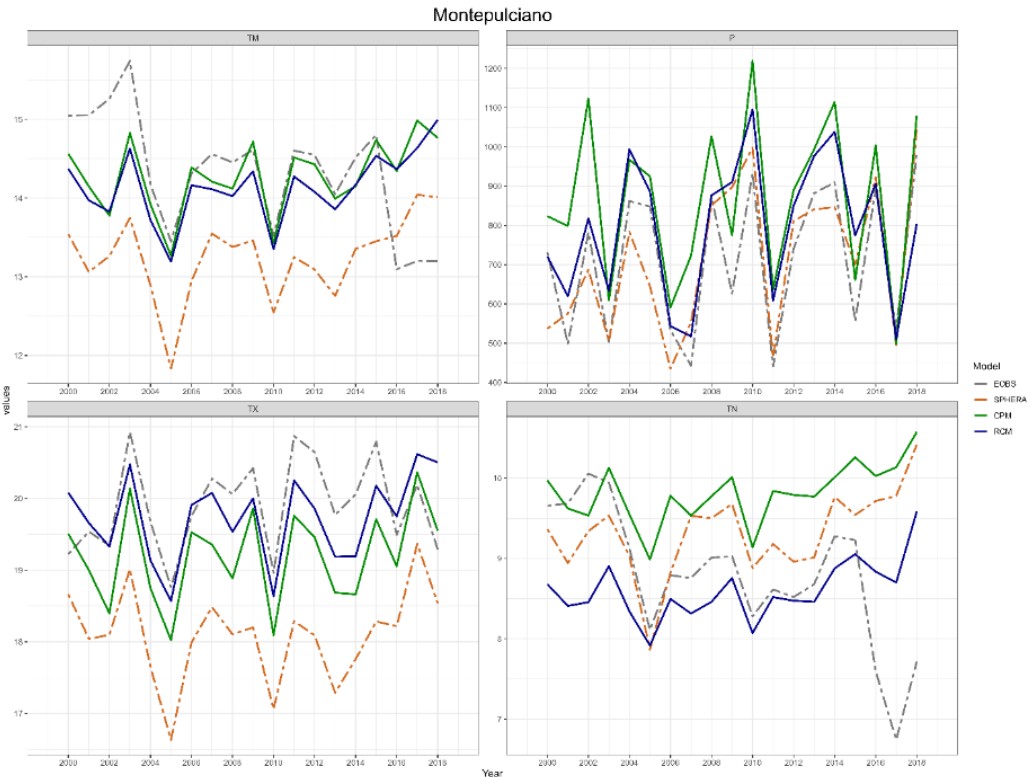

**Figure A 3: Time series of mean (TM), maximum Temperature (TX), minimum (TN) temperature and precipitation (P) over MON area for the period 2000-2018. All the time series are based on data remapped on E-OBS grid (~ 11 km).**

**Table A 2: Spearman correlation coefficient (ρ) , the root mean squared error (RMSE) between SPHERA (E-OBS) and CPM, as well as SPHERA (E-OBS) and RCM time series and the percentage differences of RMSE with the mean of the reference (SPHERA and E-OBS) (RMSE%). in the FRA and MON area.**

| FRA | | | | | | | | | |
|---|---|---|---|---|---|---|---|---|---|
| | TM | | TX | | TN | | P | | |
| | ρ | RMSE (°C) | ρ | RMSE (°C) | ρ | RMSE (°C) | ρ | RMSE (mm) | RMSE% |
| SPHERA CPM | **0.95\*** | 0.78 | **0.94\*** | 1.54 | **0.96\*** | 0.39 | **0.84\*** | 233.52 | 18.2 |
| SPHERA vs RCM | **0.95\*** | 0.38 | **0.96\*** | 1.73 | **0.91\*** | 1.37 | **0.73\*** | 415.05 | 32.4 |
| E-OBS vs CPM | **0.76\*** | 0.64 | **0.78\*** | 0.6 | **0.55\*** | 0.78 | **0.76\*** | 435.99 | 62.4 |
| E-OBS vs RCM | **0.85\*** | 0.37 | **0.82\*** | 0.43 | **0.58\*** | 0.61 | **0.77\*** | 266.65 | 38.2 |
| MON | | | | | | | | | |
| | TM | | TX | | TN | | P | | |
| | ρ | RSME (°C) | ρ | RSME (°C) | ρ | RSME (°C) | ρ | RSME (mm) | RMSE% |
| SPHERA CPM | **0.79\*** | 1.06 | **0.81\*** | 1.15 | **0.78\*** | 0.58 | **0.78\*** | 196.26 | 27.9 |
| SPHERA vs RCM | **0.86\*** | 0.91 | **0.92\*** | 1.66 | **0.77\*** | 0.77 | **0.78\*** | 133.12 | 18.9 |
| E-OBS vs CPM | 0.16 | 0.79 | **0.65\*** | 0.85 | -0.08 | 1.39 | **0.86\*** | 177.98 | 26.2 |
| E-OBS vs RCM | 0.06 | 0.83 | **0.52\*** | 0.57 | 0.04 | 0.94 | **0.8\*** | 128.03 | 18.8 |




**Table A 3: Welch's t-test between SPHERA (E-OBS) and CPM, as well as the SPHERA (E-OBS) and RCM time series in the FRA and MON.  For each variable (TM, TX, TN and P) the test statistics (t.stat), the t tabulated or critic (t.tab) for a 95% confidence interval and the degree of freedom (Dof) computed using Welch's formula are reported. Bold font and an asterisk (*) indicate the p-value <= 0.05, i.e. the rejection of the null hypothesis (h0) and a statistically significant difference between the mean value of the series.**

| FRA | | | | | | | | | | | |
|---|---|---|---|---|---|---|---|---|---|---|---|
| | TM | | | TX | | | TN | | | P | | |
| | t.stat | t.tab | Dof | t.stat | t.tab | Dof | t.stat | t.tab | Dof | t.stat | t.tab | Dof |
| SPHERA vs CPM | **4.16*** | 2.03 | 35.2 | **6.7*** | 2.03 | 34.25 | **-2.31*** | 2.03 | 35.96 | **-2.07*** | 2.03 | 35.85 |
| SPHERA vs RCM | 1.8 | 2.03 | 34.77 | **7.77*** | 2.03 | 34.83 | **-8.26*** | 2.03 | 35.59 | **-4.48*** | 2.03 | 35.47 |
| E-OBS vs CPM | **2.98*** | 2.03 | 33.1 | -1.54 | 2.03 | 35.76 | **3.84*** | 2.03 | 36 | **4.93*** | 2.04 | 29.22 |
| E-OBS vs RCM | 0.5 | 2.04 | 32.45 | -0.75 | 2.03 | 35.95 | **-2.24*** | 2.03 | 35.83 | **2.91*** | 2.04 | 32.58 |
| MON | | | | | | | | | | | |
| | TM | | | TX | | | TN | | | P | | |
| | t.stat | t.tab | Dof | t.stat | t.tab | Dof | t.stat | t.tab | Dof | t.stat | t.tab | Dof |
| SPHERA vs CPM | **6.45*** | 2.03 | 35.57 | **5.24*** | 2.03 | 35.97 | **3.38*** | 2.04 | 32.37 | **2.33*** | 2.03 | 35.69 |
| SPHERA vs RCM | **5.72*** | 2.03 | 35.03 | **8.15*** | 2.03 | 35.83 | **-4.8*** | 2.04 | 32.12 | 1.3 | 2.03 | 35.91 |
| E-OBS vs CPM | -0.24 | 2.04 | 30.12 | **-3.29*** | 2.03 | 35.99 | **4.89*** | 2.06 | 24.89 | **2.37*** | 2.03 | 35.57 |
| E-OBS vs RCM | -0.95 | 2.04 | 29.09 | -0.81 | 2.03 | 35.76 | -0.87 | 2.06 | 24.71 | 1.34 | 2.03 | 35.96 |

**Table A 4: Welch's t-test between SPHERA (E-OBS) and CPM, as well as the SPHERA (E-OBS) and RCM time series in the FRA and MON.  For each bioclimatic index the test statistics (t.stat), the t tabulated or critic (t.tab) for a 95% confidence interval and the degree of freedom (Dof) computed using Welch's formula are reported. Bold font and an asterisk (*) indicate a p-value  <= 0.05, i.e. the rejection of the null hypothesis (h0) and a statistically significant difference between the mean value of the series.**

| FRA | | | | | | | | | | | | |
|---|---|---|---|---|---|---|---|---|---|---|---|---|
| | SPHERA vs CPM | | | SPHERA vs RCM | | | E-OBS vs CPM | | | E-OBS vs RCM | | |
| Index | t.stat | t.tab | Dof | t.stat | t.tab | Dof | t.stat | t.tab | Dof | t.stat | t.tab | Dof | Index |
| BEDD (GDD) | -0.92 | 2.03 | 35.97 | -0.17 | 2.03 | 35.97 | 0.67 | 2.03 | 35.36 | 1.47 | 2.03 | 35.35 | BEDD (GDD) |
| HI (GDD) | **-4.50*** | 2.04 | 32.50 | **-4.71*** | 2.03 | 33.34 | -0.88 | 2.04 | 32.14 | -0.96 | 2.03 | 33.01 | HI (GDD) |
| WI (GDD) | **-4.48*** | 2.04 | 32.68 | **-4.13*** | 2.04 | 32.65 | **-3.25*** | 2.04 | 30.29 | **-2.89*** | 2.04 | 30.26 | WI (GDD) |
| TmVeg (°C) | **-4.59*** | 2.04 | 32.60 | **-4.17*** | 2.04 | 32.59 | **-3.28*** | 2.04 | 30.54 | **-2.85*** | 2.04 | 30.53 | TmVeg (°C) |
| TnVeg (°C) | **2.86*** | 2.03 | 32.92 | **5.35*** | 2.03 | 35.87 | -0.16 | 2.04 | 30.41 | **2.42*** | 2.03 | 34.63 | TnVeg (°C) |
| TxVeg (°C) | **-8.32*** | 2.03 | 32.82 | **-8.62*** | 2.03 | 35.95 | **-5.47*** | 2.04 | 30.10 | **-5.30*** | 2.03 | 34.76 | TxVeg (°C) |
| CNI (°C) | 0.99 | 2.03 | 33.37 | **2.29*** | 2.03 | 35.16 | -1.22 | 2.03 | 33.70 | -0.11 | 2.03 | 35.37 | CNI (°C) |
| TnRest | -0.23 | 2.03 | 35.51 | **2.69*** | 2.03 | 35.40 | **-2.53*** | 2.03 | 35.77 | 0.15 | 2.03 | 35.84 | TnRest |
| GSP (mm) | **5.55*** | 2.03 | 35.93 | **8.76*** | 2.03 | 33.94 | **-4.23*** | 2.04 | 32.17 | -1.48 | 2.03 | 35.20 | GSP (mm) |
| SprR (mm) | -0.03 | 2.03 | 36.00 | 1.92 | 2.03 | 35.18 | **-3.80*** | 2.04 | 31.84 | -1.86 | 2.03 | 34.38 | SprR (mm) |
| MON | | | | | | | | | | | | |
| | SPHERA vs CPM | | | SPHERA vs RCM | | | E-OBS vs CPM | | | E-OBS vs RCM | | |
| Index | t.stat | t.tab | Dof | t.stat | t.tab | Dof | t.stat | t.tab | Dof | t.stat | t.tab | Dof | Index |
| BEDD | **-** | 2.03 | 35.88 | **-2.13*** | 2.03 | 35.84 | 1.91 | 2.03 | 34.16 | **2.04*** | 2.03 | 34.04 | BEDD |



| | | | | | | | | | | | | | |
|---|---|---|---|---|---|---|---|---|---|---|---|---|---|
| (GDD) | 2.25* | | | | | | | | | | | | (GDD) |
| HI (GDD) | -3.31* | 2.03 | 34.11 | -3.71* | 2.03 | 35.41 | -1.37 | 2.03 | 33.35 | -1.65 | 2.03 | 34.90 | HI (GDD) |
| WI (GDD) | -5.21* | 2.03 | 34.38 | -5.66* | 2.03 | 35.53 | -2.14* | 2.03 | 36.00 | -2.37* | 2.03 | 35.56 | WI (GDD) |
| TmVeg (°C) | -5.38* | 2.03 | 34.59 | -5.79* | 2.03 | 35.61 | -2.06* | 2.03 | 35.96 | -2.24* | 2.03 | 35.38 | TmVeg (°C) |
| TnVeg (°C) | -0.54 | 2.03 | 35.91 | 2.90* | 2.03 | 35.78 | -1.35 | 2.03 | 33.90 | 1.70 | 2.03 | 33.44 | TnVeg (°C) |
| TxVeg (°C) | -5.43* | 2.03 | 35.98 | -5.36* | 2.03 | 35.06 | -3.74* | 2.03 | 35.86 | -3.57* | 2.03 | 34.60 | TxVeg (°C) |
| CNI (°C) | -1.61 | 2.03 | 33.38 | 0.98 | 2.03 | 34.58 | -3.31* | 2.03 | 34.96 | -0.92 | 2.03 | 35.70 | CNI (°C) |
| TnRest | -2.27* | 2.03 | 35.17 | -0.82 | 2.03 | 34.45 | -2.35* | 2.03 | 33.56 | -1.01 | 2.04 | 32.57 | TnRest |
| GSP (mm) | -1.05 | 2.04 | 31.29 | 2.46* | 2.03 | 35.02 | -3.06* | 2.05 | 26.93 | -0.04 | 2.03 | 35.74 | GSP (mm) |
| SprR (mm) | -2.44* | 2.05 | 27.64 | -0.44 | 2.04 | 31.33 | -2.75* | 2.04 | 32.09 | -0.95 | 2.03 | 35.18 | SprR (mm) |



**Table A 5: Sen's slope FRA, bold font, and asterisk (*) indicate a significant trend (p<=0.05)**

| FRA | TM (°C/yr) | TX (°C/yr) | TN (°C/yr) | P (mm/yr) | BEDD (GDD/yr) | HI (GDD/yr) | WI (GDD/yr) | TmVeg (°C/yr) | TnVeg (°C/yr) | TxVeg (°C/yr) | CNI (°C/yr) | TnRest (°C/yr) | GSP (mm/yr) | SprR (mm/yr) |
|---|---|---|---|---|---|---|---|---|---|---|---|---|---|---|
| E-OBS | **0.05*** | 0.05 | **0.06*** | -5.91 | **4.59*** | **14.96*** | 11.67 | **0.06*** | 0 | 0.1 | 0.09 | 0.03 | -4.77 | -1.33 |
| SPHERA | 0.04 | 0.03 | **0.04*** | 12.89 | 4.5 | 9.25 | 6.65 | 0.04 | 0.02 | 0.05 | 0.1 | 0.02 | **13.32*** | **4.57*** |
| CPM | 0.04 | 0.03 | 0.04 | 6.54 | 3.35 | 13.34 | 12.61 | 0.06 | 0.01 | 0.12 | **0.13*** | 0.05 | -1.31 | 0.7 |
| RCM | **0.05*** | 0.04 | **0.04*** | -2.14 | 4.19 | 11.51 | 11.94 | 0.06 | **0.05*** | **0.12*** | 0.12 | 0.07 | -2.41 | -0.15 |

**Table A 6: Sen's slope productivity FRA bold font, and asterisk (*) indicate a significant trend (p<=0.05)**

| FRA | Productivity (q/ha)/yr |
|---|---|
| slope | **1.28*** |



**Table A 7: Sen's slope MON, bold font, and asterisk (*) indicate a significant trend (p<=0.05)**

| MON | TM (°C/yr) | TX (°C/yr) | TN (°C/yr) | P (mm/yr) | BEDD (GDD/yr) | HI (GDD/yr) | WI (GDD/yr) | TmVeg (°C/yr) | TnVeg (°C/yr) | TxVeg (°C/yr) | CNI (°C/yr) | TnRest (°C/yr) | GSP (mm/yr) | SprR (mm/yr) |
|---|---|---|---|---|---|---|---|---|---|---|---|---|---|---|
| E-OBS | **-0.07*** | 0.04 | **-0.11*** | 8.64 | **-7.89*** | 1.23 | **-17.42*** | **-0.08*** | -0.09 | 0.07 | -0.07 | 0.03 | 4.38 | 0.07 |
| SPHERA | 0.03 | 0.01 | **0.03*** | **19.47*** | 2.94 | 5.05 | 7.22 | 0.03 | **0.1*** | **-0.08*** | **0.12*** | 0 | **10.36*** | 0.99 |
| CPM | 0.03 | 0.02 | **0.03*** | 5.28 | 2.42 | 6.84 | 3.68 | 0.02 | **0.05*** | **0.05*** | 0.15 | 0 | 0.74 | 1 |
| RCM | 0.04 | 0.03 | **0.03*** | 6.28 | 1.2 | 10.5 | 9.31 | 0.04 | **0.06*** | 0.01 | **0.11*** | 0.06 | -0.08 | 0.34 |

**Table A 8: Sen's slope productivity MON, bold font, and asterisk (*) indicate a significant trend (p<=0.05)**

| MON | Productivity (q/ha)/yr |
|---|---|
| slope | 0.43 |




**Table A 9: ranking of the maximum variance (%) explained for each dataset for each consortium, with the indication of type of**

**method used (SR: single regression, MR multi-regression.)**

| FRA | | | MON | | |
|---|---|---|---|---|---|
| Model | var.value % | type | Model | var.value % | type |
| RCM | 64 % | MR | CPM | 45 % | MR |
| SPHERA | 56 % | MR | E-OBS | 44 % | SR |
| CPM | 48 % | MR | SPHERA | 42 % | MR |
| E-OBS | 42 % | SR | CPM | 34 % | SR |
| SPHERA | 36 % | SR | RCM | 32 % | SR |
| E-OBS | 35 % | MR | E-OBS | 32 % | MR |
| RCM | 35 % | SR | RCM | 29 % | MR |
| CPM | 34 % | SR | SPHERA | 21 % | SR |