# Peer review of "Using a convection-permitting climate model to assess wine grape productivity: two case studies in Italy"

_EGUsphere, 2024_

## Referee Comment (RC2)

**Manuscript number**: egusphere-2024-941
**Title**: Using a convection-permitting climate model to predict wine grape productivity: two case studies in Italy
**Authors**: Laura T. Massano, Giorgia Fosser, Marco Gaetani, Cécile Caillaud

**Summary**

The manuscript explores the impact of convection-permitting climate model data on assessing wine grape productivity. The analysis utilizes observed wine productivity data from two consortia located in Italy. First, the climate data are compared with both meteorological observations and reanalyses to evaluate their quality. Second, single and multiple regression analyses are conducted to investigate the potential of climate-related indices to "predict" wine productivity. The results show correlations between bioclimatic indices (in particular temperature-based indices) and wine productivity, suggesting their potential use in assessing future changes.

**General comments:**

The manuscript is well written, data and methods are described adequately; the topic treated adheres to the journal's scope. In my opinion, the manuscript would benefit from a more extensive discussion comparing the findings to previous similar studies or applications of CPM climate data in agriculture. Additionally, providing in-depth comments on the potential impact of this research, particularly its implications for future grape productivity, would underscore the relevance of the study.

My general assessment is that the manuscript doesn't have any relevant flaws that prevents its publication. My recommendation is to accept the paper, provided that the specific remarks below are addressed.

English is not my native language and I have no comments on it.

**TITLE**

I suggest to modify the title, e.g., from "Using a convection-permitting climate model to predict wine ..." to "Using a convection-permitting climate model to **assess/estimate** wine grape productivity: two case studies in Italy".

In fact, the authors make use of single/multiple regression to explain the variance of wine productivity data and not to predict them.

**ABSTRACT**

**General comment**

lines 7-9: "Viticulture is tied to climate, it influences the suitability of an area, yield and quality of wine grapes. Therefore, traditional wine-growing regions could be threatened by a changing climate. Italy is at-risk being part of the Mediterranean climatic hotspot and judged in 2022 the second-largest exporter of wine worldwide."I suggest removing this sentence as the opening statement. It is redundant in the Abstract but appropriate for the Introduction section.

**Specific comments**

Line 13: "and both **the** Regional and **the** Convection-permitting..." which models? If not detailing them, use the article "a" instead of "the".

Line 14: "The potential of CPM**s**", which CPMs? One CPM or multiple CPMs? Remove the "s"

line 17: "of CPM**,** became" remove the comma "," which currently lies between the subject and the verb

line 17: "of CPM**,** *became*" mixed use of present and past. Please change to "becomes"

**1. INTRODUCTION**

**General comment**

The Introduction is adequate and presents the state-of-the-art and the innovative approach of the research (from line 50 to the end of the section). As stated previously and for the reasons claimed by the authors in the Introduction (see lines 65-66: "Single and multiple regression approaches are used to determine the extent to which bioclimatic indices *can explain changes* in wine grape productivity at the local scale"), I suggest modifying the title by removing the word 'predict' and replacing it with a more generic term such as 'assess' or 'estimate.' In fact, predicting implies providing the accuracy of the predictions, including associated errors and uncertainties, rather than just the R-squared value of the regression.

**Specific comments**

line 21: I would start the Section with the first sentence of the Abstract, which I found redundant in that context.

Line 21-22: "Wine-growing has a strong socio-economic impact and is one of the principal agricultural economic activities in Italy, that in 2022..." I would change to "Wine-growing has ... in Italy**. I**n 2022..."

line 22: "world's leading wine producer (49.8 million hl)" Can the authors provide any reference to support this statement?

line 22: "and second" change to "and the second"

line 27: "than the global average (Bernetti et al., 2012;.." Can you provide a more climatic-sound references to support such sentence? Further, these two references could be moved below (i.e., line 28)

line 31: "when frost events are still frequent" I think the word "still" can be removed to streamline the sentence

line 33: "are expected to experience important shifts in viticulture suitability that can consequently causes a decline in production" "causes" or "cause"? What is the subject? "shifts" or suitability?

Line 26: "developed" change to "computed" or something else

line 48: modify ") ("

line 65: "are used to determine the extent to which" change to "are used to determine to which extent"?

**2. Data and Methods**
**General comment**

I think it should be explained better why the authors used Aladin/Arome model outputs rather than SPHERA or other regional CP reanalyses. Can the authors stress the differences in the experimental design of numerical simulations? As stated in the manuscript, Aladin model is fed by ERA-Interim data, whereas SPHERA is fed by ERA5. Which is the difference? SPHERA are started frequently (once a day?) and receive boundary conditions every hour. What about the Aladin/Arome numerical architecture? The authors should give the audience a taste of the differences without delving into the suggested bibliography.

**Specific comments**

Figure 1: In both the digital and hardcopy versions of the manuscript, the geographical locations of the two consortia are difficult to discern from the images. Could the authors provide larger images and/or magnify the map of Italy?

Line 102: "at regional level between 1994 and 2000; at national scale while" change to "at the regional level between 1994 and 2000; at the national scale while"

line 103: "at national scale while from 2000 to 2005" sentence not clear. Perhaps just remove "while"

line 112: remove the acronym "NMHSs" since it is used only once

line 124: "SPHERA reanalysis" change to "The SPHERA reanalysis" or just "SPHERA"

line 124: SPHERA is validated against a gridded dataset made of independent rain-gauges. ERA5 data are used as a benchmark. Please specify better

line 138:

line 139: "." is missing at the end of the text

line 163: "but also take" change to "but also takes"

line 199: "Tests performed to investigate…are not impacted by the resolution chosen for the remapping (not shown)" I think the authors should give more details about the tests performed. At least they should say whether the tests were performed on the remapping strategy (i.e., the algorithm) or on the resolution (i.e., the final grid spacing). Please expand this point and give some details

line 205: "weighing" please replace with "weighting"

line 214: "SPHERA and E-OBS time series together provide a range within which the CPM and the RCM time series are expected to fall, similar to a 'confidence interval'." I disagree; can the authors support this statement by providing evidence?

Line 2017: I don't get why E-OBS are used within the parenthesis

Line 221: too many "()"

**3. Results**
**General comment**

I don't see why many Figures/Tables that are commented in this Section are taken from the Appendix (e.g., lines 255, 258, 261, 265, 267, 277, 298, 306, 316). This doesn't help the readability of the manuscript. I encourage the authors to rethink this section. For example, it should start from line 269 "Figure 2 and Figure 3 show the ten bioclimatic…" and the first

sentence "The precipitation and temperature time series of both..." could be moved elsewhere in the manuscript (Discussion or Appendix). Alternatively, some tables/plots shown in the Appendix could be streamlined (e.g., Table A 4 which has many columns and rows) or simply removed and their content moved into the text (e.g., Table A 6, Table A 8). Table 2 is hard/difficult to read. I wonder whether a plot could help its readability. I suggest the authors to reconsider it. If they decide to keep it, I suggest to remove the RMSE column, since it is not commented in the text. Further, it is shown the RMSE% which is more informative since the ranges of bioclimatic indexes are very different.

**Specific comments**
line 276: "( Table 2)" remove empty space
line 296: "CNI with model simulations" change to "CNI with model **climate** simulations" or simply "CNI with climate simulations", in fact, SPHERA is a model simulation too
lines 302-306: I would like to see this paragraph in the Discussion section, where it is more pertinent
lines 325-330: as above
Table 3: Can the authors discuss why, in MON, the variance of E-OBS is 44% for the SR case and 32% for MR? Is it related to the poor quality of E-OBS data (in MON) as argued previously? This happens also for RCM although to a smaller extent (32%→29%). I would like to see a plausible explanation in the Discussion section

**4. Discussion and conclusion**
**General comment**
I think this section lacks a critical review of the results found in comparison with previous studies. It looks like a summary of the manuscript. If no or few previous studies are found, it should stressed the novelty of the study and highlighted the potential and limits of CP model data in assessing productivity.
Further, the main advantage of using CPM data is the removal of any parameterisation to model convection processes. Indeed, it is well known they provide more accurate precipitation estimates than RCM data (e.g., lines 53-55). However, you found that wine productivity is mostly related to temperature-based bioclimatic indexes rather than precipitation-based one. Do the authors have any comment on it?
**Specific comments**
line 401: "This could be link" replace with "This could be linked"

**Data availability**
**Author contribution**
**Competing interests**
**Acknowledgments**
**References**
line 573: incomplete reference
**Appendix A**
Figure A2 provided in poor quality when on hardcopy

**Studies cited above:**
*none*

---

## Author Comment (AC1)

**Preliminary remarks**

As suggested by Anonymous Referee #2, the tittle of the manuscript has been modified in "Using a convection-permitting climate model to assess wine grape productivity: two case studies in Italy" to better reflect the aim of the paper.

In addressing the reviewers' comments, we decided to modify the structure of the article. Different sections of the article have been moved and streamlined, to account for the suggestions of both reviewers and improve readability and the fluency of the text.

The most relevant changes are reported below:

As suggested by Reviewer #1, the Abstract has been rephrased to enhance its precision and focus.

2. Data and Methodology: Subsection "2.6 Trend Analysis" has been added, and the order of the two subsections, "2.4 Bioclimatic indices" and "2.5 Validation of climate simulations and calculation of bioclimatic indices" has been reversed. Section "2.5 Validation of climate simulations and calculation of bioclimatic indices" has been renamed "2.5 Validation of climate simulations" to better reflect its content. The revised order is now as follows: "2.4 Validation of climate simulations "; "2.5 Bioclimatic indices"; "2.6 Trend Analysis".

3. Results: Subsection "3.1 Validation of the climate simulations" has been streamlined, Table 2 was replaced by two new figures, Figure 6 and 7, to increase the readability. Figure A2 and A3 showing the time series of mean (TM), maximum Temperature (TX), minimum (TN) temperature and precipitation (P) over FRA and MON area have been moved to the main text before Figures 2 and 3showing the time series of the bioclimatic indices from 2000-2018 averaged on the FRA and MON consortium area.

At last, the "4 Discussion and Conclusion" section has undergone a comprehensive revision and expansion.

Further details are set out in our point-by-point response (in blue), which can be found below.

**Response to RC-1**

Manuscript number: egusphere-2024-941

Title: Using a convection-permitting climate model to predict wine grape productivity: two case studies in Italy

Authors: Laura T. Massano, Giorgia Fosser, Marco Gaetani, Cécile Caillaud

**General comments:**

- This study investigates correlations between wine productivity and bioclimatic indices, calculated based on observations, reanalysis data, RCM, and CPM. Temperature and precipitation-based indices explain the changes in wine productivity in two consortia. The simulation output from climate models is validated at a local scale and is expected to predict

productivity using highly correlated indices. It provides a valuable contribution to understanding the variability impact of the climate on viticulture.

For this reason, this work should be published after several improvements are made. The written text should be proof checked. The presentation of the work can be much improved regarding the description of the dataset and interpretation of the results (see comments below).

We thank the Reviewer for taking the time to revise our manuscript and providing constructive comments, which helped in improving the paper. Please find below our point-by-point response, highlighted in blue.

**Specific comments:**

- The abstract can be sharp and more specific. Please specify "some bioclimate indices" in line 15. It should also include a discussion on two cases, highlighting why it is necessary to use CPM.

Thank you for the comment. We modified the abstract according to the suggestion as follow: "[…]. Temperature and precipitation-based bioclimatic indices are computed using the observational dataset E-OBS, the high-resolution climate reanalysis product SPHERA, the regional climate model CNRM-ALADIN and the km-scale convection-permitting climate model CNRM-AROME. The multi-regression method outperforms the single regression, systematically enhancing the ability of bioclimatic indices to explain productivity variability. The results show that productivity is strongly tied with temperature-based bioclimatic indices in the area of "Consorzio per la tutela del Franciacorta" in northern Italy, while for the "Consorzio del Vino Nobile di Montepulciano" area in central Italy both temperature and precipitation-based indices are relevant. Climate models, providing similar results as E-OBS and SPHERA, appear to be a useful tool to explain productivity variance. In particular, the added value of convection-permitting resolution is evident when precipitation-based indices are considered. […]"

- a good explanation of frost risks on line 31.

We rephrase the paragraph to clarify:

"Since temperature is the primary driver for the phenological phases (Fraga et al., 2016), a warmer climate may lead to a shorter growing cycle and an earlier onset of phenological phases, which would increase frost-related risk (Lamichhane, 2021; Trought et al., 1999). In fact, budburst is the most vulnerable phase to frost in the vine growing cycle, and an earlier budburst in spring would increase the exposure of the vine to late frost events."

- line 33, please explain "important shifts" in "important shifts in viticulture suitability".

We have clarified by rephrasing the paragraph:

"Furthermore, climate conditions typical of traditional wine-producing regions, such as Douro in Portugal, La Rioja in Spain, Bordeaux in France, and Tuscany in Italy, are expected to shift northwards or at higher altitude, and this modifications in viticulture suitability may consequently cause a decline in production (Adão et al., 2023; Rafique et al., 2023; Sgubin et al., 2023; Tóth and Végvári, 2016)."

- line 64, "the driving RCM simulation" what drives RCM simulation

The RCM simulation is used to drive the CPM simulation and is in turn driven by ERA-Interim reanalysis. This aspect is elaborated in the methodological section. Sentence at line 64 has been rephrased to clarify:

"The CPM performance is validated against climate observations and a km-scale reanalysis product. Furthermore, the added value of the higher resolution is assessed by comparing the CPM to an RCM simulation.".

- Figure 1 does not show clearly the scale, north direction, and regions (FRA, LOM, MON). The zooming-in parts are not clearly shown in the map.

Thanks for the suggestion, we have replaced the figure with a more informative one.

- line 95, okay, it is a fair assumption

Thank you.

- line 98, "thus contextualise this work within the broader framework of previous studies", what specific framework? is it necessary to mention this?

Thank you for pointing this out. The sentence has been modified and now reads: "To assess the consistency of productivity data between local and regional scales, the productivity at the local scales (FRA and MON) is compared with productivity at regional scale provided by the Italian National Institute of Statistics (ISTAT)."

- line 107, "the Welch's t-test proves that both consortium distributions are part of the regional population", why this test can prove this conclusion here. please simply explain. What are the variables used in the table. please explain and not directly use terms from the code.

The application of the Welch's t-test is explained in the text:

"In addition, the application of a Welch's t-test, designed to assess whether two samples are extracted from the same population, proves that the productivity distributions of both consortia are consistent with the respective regional productivity distributions (Table A 1 and Figure A 1)."

The header of Table A1 has been modified:

"Table A 1: Results of Welch's t-test applied to regional and consortia productivity data: t statistics (t), reference value for t (tref), degrees of freedom (DoF) for the t-test based on the number of observations computed according to the Welch's equation for effective degrees of freedom (Welch, 1947) are displayed. Values of t lower than $t_{ref}$ indicate that consortium and regional productivity samples comes from the same population, at 95% level of confidence. In the last column: temporal correlation coefficient (r) computed between consortium and regional productivity data. Asterisks (*) indicate statistically significant correlations ($p <= 0.05$)."

In addition, the headers of Tables A3 and A4 have been improved:

"Table A 3: Results of Welch's t-test applied to mean (TM), maximum (TX) and minimum (TN) temperatures and precipitation (P) from E-OBS, SPHERA, RCM and CPM datasets, for FRA and MON: t statistics (t), reference value for t (tref), degrees of freedom (DoF) for the t-test based

on the number of observations computed according to the Welch's equation for effective degrees of freedom (Welch, 1947) are displayed. Values of t higher than tref indicate that the samples from climate model simulations and the reference datasets come from different populations, at 95% level of confidence. Asterisks (*) indicate the means showing statistically significant differences."

"Table A 4: Results of Welch's t-test applied to the bioclimatic indices from E-OBS, SPHERA, RCM and CPM datasets, for FRA and MON: t statistics (*t*), reference value for *t* (tref), degrees of freedom (DoF) for the t-test based on the number of observations computed according to the Welch's equation for effective degrees of freedom (Welch, 1947) are displayed. Values of t higher than tref indicate that the samples from climate model simulations and the reference datasets come from different populations, at 95% level of confidence. Asterisks (*) indicate the means showing statistically significant differences."

- line 154, "more than 3000 degrees per day", please explain if the unit is correct. Also the same unit used below

We have corrected the text, the unit is degree days

- line 160, please explain the unit

The correct unit is degree days.

- line 173, by "occurrence", do you count the frequency of frost days per year?

No, we don't count the frequency of the frost days. The index indicates the minimum temperature the vines are exposed during the vegetative period. We have rephrased for clarity.

"Minimum temperature during vegetative period (TnVeg), which is the minimum temperature recorded during the vegetative period (1st April to 31st October). This index is important to assess whether the vines are exposed to low temperature or even to spring frosts that pose a significant risk to viticultural practices and production. The damage threshold is fixed at -2 °C (Sgubin et al., 2018)."

- line 200, The remapping strategy is the key to the workflow. Please explain the remapping strategy (average, weighted average, (fraction-) interpolation). How are datasets with different resolutions (OBS: 11 km, reanalysis: 2.2 km, RCM: 12.5 km, CPM: 2.5 km) interpolated onto the same grid? What happens when grids from different datasets do not overlap?

The reviewer is right, the remapping strategy is an important step for the analysis. Because all the analysed datasets have different grids, we remapped all to the coarsest one (E-OBS at ~11 km), using a conservative remapping. The paragraph has been rephrased for clarification:

"To compare datasets with different horizontal resolutions on equal terms (Berg et al. 2013), observations, reanalysis and model simulations are conservatively remapped on a common grid, i.e. the E-OBS regular grid at ~11 km, the coarsest among all. Tests performed to assess the impact of upscaling SPHERA and CPM at a coarser resolution showed no significant changes in the results (not shown)."

- line 200-210, what are the temporal resolutions of the datasets? I expect the value of accumulative variables to depend on the temporal resolution.

The analysis is performed at daily time scale. E-OBS data are provided at daily resolution, while SPHERA, CPM and RCM data, originally at hourly resolution, are aggregated at the daily time scale to match E-OBS data. We clarify this aspect in Sections 2.2 and 2.3.

- the CNI indices looks like an accumulative value, but the suitable class range is within 12 -18 degrees.

The CNI is not an accumulative value but is an average of the minimum air temperatures of September. We rephrased for clarity as: "Cool Night Index (CNI), defined as the average of minimum air temperatures during the month of September".

- line 121, what do you mean by "common period".

We have modified the sentence, which now reads "CPM performance is evaluated for the period 2000-2018", since the previous paragraph clarifies that "The analysis focuses on the 19 years from 2000 to 2018 that is the longest period available for RCM and CPM simulations and shared with E-OBS, SPHERA climate data and FRA and MON productivity data."

- line 215, "SPHERA and E-OBS time series together provide a range within which the CPM and the RCM time series are expected to fall, similar to a 'confidence interval'." I understand that observations and reanalysis data are taken as a reference. I don't understand why observations and reanalysis data "provide a range" since observations and reanalysis data are not statistical tests. i don't see why CPM and RCM should fall into this range. Please rephrase.

The sentence was rephased for clarity.

"The comparison between climate model simulations and the reference datasets is carried out by computing the Spearman correlation, the Root Mean Square Error (RMSE) and the Normalised Root Mean Square Error (NRMSE) with respect to the range of values, i.e. the maximum value of the variable considered ($y_{max}$) minus the minimum value ($y_{min}$), for the reference datasets (SPHERA and E-OBS)."

- line 233, please rephrase the sentence.

We have rephrased it to clarify the objective of the statistical tools we are using:

"In particular, the Spearman correlation coefficient is used to assess the ability of the climate models in reproducing the climate variability of the reference datasets, while RMSE and NRMSE provides a measure of the climate models biases. Moreover, the statistical significance of the model biases is assessed by applying a Welch's two-tailed t-test (Welch, 1938), with a 95% level of confidence."

- line 236, "The best subsets regression technique" what do you mean by subsets regression technique, why it is the best - the multi-regression part can be improved by specifically describing how the bioclimatic data is being processed.

The "best subset selection" refers to the approach described by James et al., 2013 (*James G, Witten D, Hastie T, Tibshirani R. An Introduction to Statistical Learning: With Applications in R. 1st ed. 2013, Corr. 7th printing 2017 edition. Springer; 2013)*. We utilize the "best subset selection" approach when constructing the regression model. This approach aims at identifying the subset of predictors (the bioclimatic indices) that most effectively predicts the

predictand (productivity) by considering all potential combinations of independent variables. The method was already employed in a similar study by Massano et al (2023), showing good results. The method is described in the revised version of the manuscript as follow:

"The best-subset selection approach, implemented by James et al. (2013), is used to optimise the prediction of productivity, as in Massano et al. (2023). This approach seeks the subset of predictors, i.e., the bioclimatic indices in this case, that most accurately predicts the predictand, i.e. the productivity, by examining all feasible predictor combinations and thus selecting the one minimising the error in the prediction. This is achieved by utilising the k-fold cross-validation method. The k-fold cross validation method is employed to identify the optimal model (Kassambara, 2017). This method performs cross-validation by randomly dividing the data into k subsets of approximately equal size, with k typically set to 5 or 10 (here k = 5 is used). One of the folds serves as test set and the remaining as training set."

- line 249, could you explain "the variance explained by the MR model"? variance of what, what does "explain" the variance mean? you mean the model is able to predict

The MR model is used to predict the observed productivity. Its performance is assessed by computing the correlation between observed productivity and the productivity predicted by the MR model. The variance of observed productivity explained by the predicted productivity is estimated as the determination coefficient, i.e. the squared of the correlation coefficient. The sentence has been modified as follows:

"The so-optimised MR model (productivity = a1*index1 + a2*index2 + a3*index3 + ..., with "indexn" indicating the selected bioclimatic index) is then used to predict the productivity and the Pearson correlation between predicted and observed productivity is calculated.

Following Massano et al. (2023), the comparison between the SR and MR methods is performed in terms of the productivity variance explained by the prediction, estimated by computing the coefficient of determination, i.e. the square of the correlation coefficient."

- the variance of the productivity data to some degree (percentage). The terms here are vague. This applies to similar terms used later on.

This study is based on the analysis of time series by means of correlation and regression analysis. Therefore, we refer in the manuscript to the variance of the distributions of the analysed time series explained by the statistical models used. This applies to climate variables, bioclimatic indices and productivity data. This has been clarified by improving the description of the methodology.

- line 253- 255, sorry i couldn't understand what you mean here, "evaluate biases"; "could lead to biases"

Section 3.1 has been rearranged and line from 253 to 269 has been streamlined to improve readability. We have clarified that the existing bias in the climatic variables, i.e. temperature and precipitation, could result in biases in the bioclimatic indices (that are based on temperature and precipitation).

Now the first paragraph (that was lines 253-269) reads:

"Prior to the computation of the bioclimatic indices, the precipitation and temperature fields in both consortia (FRA and MON) are analysed to assess the potential biases, which could impact on the temperature and precipitation-based bioclimatic indices. Figure 2 for FRA and Figure 3 for MON show the precipitation (P) and temperature (TM: mean temperature, TX: max temperature and TN: min temperature) time series of E-OBS, SPHERA, RCM and CPM for the period 2000-2018. In general, both RCM and CPM well reproduce SPHERA temporal variability, as also confirmed by the high and significant correlations for all the climate variables in both consortia (Table A 2). Nevertheless, both climate models tend to overestimate mean and maximum temperature while underestimating minimum temperature, as reflected by the statistical differences in mean values (Table A 3). Both climate models, and especially the RCM, underestimate precipitation in FRA, while the CPM tends to overestimate it in MON. Precipitation in MON is slightly overestimated also by the RCM. In FRA In FRA, RCM is closer to E-OBS mean values than CPM (Table A3). However, in MON, E-OBS minimum temperature time series shows a strong decrease of almost 2°C between 2015 and 2018 (Figure 3), which is not observed in any models nor SPHERA. Further investigations revealed that this temperature decline is observed throughout the entire TOS and is inconsistent with other observational records (not shown). This E-OBS misrepresentation of the temperature field has a subsequent effect on the mean and minimum temperature time series (Figure 3), the temporal correlations (Table A 2), and is likely to be reflected in the temperature-based bioclimatic indices in TOS region, and at local scale in MON."

- line 255- 257, what do you mean "common period", are these variables different from your "ten indices"? if it is the statistics over the whole period, please specify that.

Section 3.1 has been streamlined and this sentence has been removed. Please refer to the response to your comment on lines 253- 255 for further details.

- line 255 - 257, since the validation part is important to show the validity of the climate model, why not put it in the main text?

We agree with the reviewer, and we have modified section 3.1 and included the figures that were previously in supplementary. Please see our response to your comment on lines 253-255.

- line 259, "Further investigations highlighted that this temperature fall affects the entire TOS" what do you mean this temperature fall affects the entire TOS? does this decreasing temperature have an influence on the TOS region? please rephrase, make it clear that the indices in the TOS region are influenced by this.

The sentence has been rephrased. Please refer to the response to your comment on lines 253-255 for further details.

- line 264-269, I suggest moving this part before the underestimated temperature part.

This paragraph has been rearranged. Please refer to the response to your comment on lines 253-255 for further details.

- line 265, "reproduce the same variability", high correlations do not mean "variability". please rephrase.

We mean that high correlations indicate that the variables show high temporal co-variability. The sentence has been rephrased, please refer to the response to the comment on lines 253-255 for further details.

- line 266, Table A3, again What are the variables used in the table. please explain and not directly used the term from the code.

The header of Table A3 has been modified accordingly to the Reviewer's comment on line 107.

- line 268, "The Welch's t-test confirmed that E-OBS is closer in mean value to RCM than CPM simulations." please rephrase.

We have modified the text (please see the response to your comment on lines 253-255) as well as the caption of Table A3 for clarity.

- line 297, i don't understand what you mean by "the long-term component."

The time series analysed in the paper can be decomposed into a long-term (low-frequency) component and an interannual variability (high-frequency) component. In the presence of trends, part of the correlation is associated with the trend. This is now clarified as follow:

"Nevertheless, some of these bioclimatic indices (i.e. BEDD for E-OBS and CNI for CPM) as well as the FRA productivity show significant trends (Table A 5 and Table A 7), thus these significant correlations may depend on the long-term variability (i.e. the trend) rather than on the interannual variability.".

- Table A5, Sen's slope needs explanation to the reader. how is the slope calculated? In Table A6, why is only productivity from FRA shown?

Table A5, A6, A7, A8 have been modified according to the Reviewer's suggestion. The Sen's slope of the climatic variable and bioclimatic indices is now displayed in Tables A5 and A6, for FRA and MON respectively. Table A8 now display the Sen's slope for both FRA and MON productivity (previously presented in Tables A6 and A8).

We have added a sentence in the Data and Method section clarifying the use of Sen's slope: "The Sen's slope estimator calculates the rate of change over time of a variable by taking the median of the slopes of all linear regressions between points pairs (Kh Aswad et al., 2020)."

We have improved the tables' captions by adding the Sen's slope definition presented in the Data and Method section.

"Table A 5: Sen's slope estimator, a statistical measure to evaluate the magnitude of the trend, for FRA area. Asterisk (*) indicate a significant trend ($p<=0.05$)"

"Table A 6: Sen's slope estimator, a statistical measure to evaluate the magnitude of the trend, for MON area. Asterisk (*) indicate a significant trend ($p<=0.05$)"

"Table A 7: Sen's slope of the productivity in FRA and MON. Sen's slope is a statistical measure used to calculate the rate of change in a variable over time, based on the Sen's estimator. Asterisk (*) indicate a significant trend ($p<=0.05$)"

- line 298-299, "RCM presents a statistically significant outcome also for TnRest". This statement seems to be a bit vague.

The sentence has been improved as follows:

"RCM presents a statistically significant and positive correlation also between productivity and TnRest, which does not show trend over the period 2000-2018, suggesting that TnRest variability has a role in controlling productivity at the interannual time scale".

- line 302, could you specify what resolution you used for "the regional scale", just for the comparison.

The sentence refers to the results by Massano et al. (2023), who applied the same methodology to predict the observed productivity aggregated at the regional scale, using E-OBS data. The sentence has been rephrased and moved to the Discussion section: "[...] These results, which are obtained at the local scale using data from wine consortia, complement and expand the previous study conducted at the regional scale by Massano et al. (2023) using ISTAT productivity data and E-OBS (v26, resolution ~11 km) climate data."

- In FRA (Table 3, line 342), could you explain why MR picks up GSP, a precipitation-based index, in RCM instead of CPM?

We thank the Reviewer for pointing this out, this comment gives us the opportunity to clarify an important aspect of our analysis. We first highlight that the MR is a statistical method that identifies the combination of indices that minimises the error without any consideration of the physical meaning of the selection. GSP from RCM shows high and significant correlation with GSP from CPM and SPHERA, as well as mean values do not show any statistically significant differences. It is surprising indeed that MR picks GSP as a predictor from RCM only. We cannot provide any physical explanation for this behaviour, and we speculate that this is just a statistical artifact, resulting from the selection of an additional predictor, which only adds a residual fraction to the variance of the observed productivity explained by the MR model. To prove this we assess the relative importance of GSP as a predictor by analysing the standardized beta coefficients of the MR in FRA for RCM (Dodge, Y. (2008). The Concise Encyclopedia of Statistics. In The Concise Encyclopedia of Statistics. Springer New York. https://doi.org/10.1007/978-0-387-32833-1). The standardised beta coefficients are expressed in standard deviations, which facilitates comparison of variables. A standardised beta coefficient enables the comparison of the strength of the effect of each individual independent variable on the dependent variable. The greater the absolute value of the beta coefficient, the more pronounced the effect. Table R1 shows that, among the predictors selected by the MR model, GSP exhibits the lowest coefficient, indicating that it had the least impact on the variance of productivity explained by the MR model. The impact on the variance in FRA explained by RCM is estimated by comparing the productivity predicted by the MR model with and without using GSP as a predictor. Results show that including the GSP as a predictor in the MR model produce a correlation coefficient of rho = 0.8 between the observed and predicted yields. When the GSP is omitted, the correlation coefficient slightly decreases to rho = 0.73, a reduction of just 0.07. Furthermore, the model with the GSP accounts for 64% of the variance, while the model without it explains 53%, showing only a modest difference in explanatory power. In conclusion, even though GSP is selected by the MR model for RCM only, it is also the

least relevant predictor. This could explain why it is not selected by the MR model for CPM and SPHERA, despite the high correlation between the series.

Table R1: Standardized coefficients of the MR model for RCM in FRA; the accompanying bar plot presents the coefficients in a visual format, facilitating comparison.

| Intercept | 0.00000 | Predictor Importance |
|-----------|---------|----------------------|
| BEDD | 0.58036 | |
| CNI | 0.49247 | |
| GSP | 0.39856 | |

We have added the following paragraph in the manuscript:

"The selection of GSP for the RCM is unexpected. Indeed, although GSP from the RCM shows high and significant correlations with both the CPM (not shown) and SPHERA (Figure 6), it is not selected by the MR model for the CPM and SPHERA. The comparison between the standardised beta coefficients (ref. Dodge 2008) of the MR model for RCM in FRA shows that GSP has the least impact on the explained variance of the observed productivity, suggesting that the selection of GSP for the RCM only might be an artifact of the statistical model."

- line 379, "in a small number of cases". do you mean a small number of indices?

We actually mean "indices". Modified as follow: "Overall, the single regression exhibits high correlation coefficients, but statistically significant results are only found for a small number of indices at the 95% confidence level"

- It would be beneficial if the authors could include a discussion on future

Following the reviewer' suggestion, we add a comment on this in the Discussion and Conclusion section.

"Nevertheless, in a changing climate, with precipitation frequency and intensity expected to change (Tramblay and Somot, 2018; Zittis et al., 2021), the relevance of precipitation, along with precipitation-based bioclimatic indices, for grape productivity might increase and in turn the use of CPM might become crucial."

**Technical corrections:**

- line 7, "…, it influence the suitability…", please clarify "it" and rephrase the sentence.

The Abstract (line 7 to 19) has been rewritten to enhance its focus and alignment with the paper's aim, following the comments from both reviewers. The new abstract is the following:

"The article explores the potential use of climate models to reproduce wine grape productivity at local scale in Italy. To this end, both single and multiple regression approaches are used to link productivity data provided by two Italian wine consortia with bioclimatic indices. Temperature and precipitation-based bioclimatic indices are computed using the observational dataset E-OBS, the high-resolution climate reanalysis product SPHERA, the regional climate model CNRM-ALADIN and the km-scale convection-permitting climate model CNRM-AROME. The multiple regression method outperforms the single regression, systematically enhancing the ability of bioclimatic indices to explain productivity variability. The results show that productivity is strongly tied with temperature-based bioclimatic indices in the area of "Consorzio per la tutela del Franciacorta" in northern Italy, while for the "Consorzio del Vino Nobile di Montepulciano" area in central Italy both temperature and precipitation-based indices are relevant. Climate models, providing similar results as E-OBS and SPHERA, appear to be a useful tool to explain productivity variance. In particular, the added value of convection-permitting resolution becomes evident when precipitation-based indices are considered. This assessment shows windows of opportunity for using climate models, especially at convection-permitting scale, to investigate future climate change impact on wine production."

Line 7 to 9 were rephrased and moved to the Introduction section where are more relevant. Now the introduction starts with:

"Viticulture is tied to climate, that influences the suitability of an area, the yield and quality of wine grapes.  The wine industry has a significant socio-economic influence and is a key agricultural sector in Italy. In 2022, Italy was the world's leading wine producer (49.8 million hl), and the second largest wine exporter, with a value of 7.8 billion euros (OIV, 2023). Over the coming decades, the wine sector is expected to be affected by climate change, especially in Italy that is part of the Mediterranean climatic hotspot (Tuel and Eltahir, 2020), where the impact of climate change is expected to be more severe than the global average."

- line 8, "at risk" no hyphen

Thank you we have modified it, please refer to the response to line 7 comment for further details.

- line 12, delete "by"

We have done it, thank you. Please refer to the response to line 7 comment for further details.

- line 16, "…, however, .." split the sentence.

We have modified it, please refer to the response to line 7 comment for further details.

- line 50, remove "the" "the wine consortia"

We have done it, thank you.

- line 53, "Thanks to" informal

Thank you, we have rephrased it.

- line 53, "without the need for parameterisation," this is inaccurate. most of the parameterizations are turned on in CPM models.

Thank you, we have rephrased the sentence: "Due to their high spatial resolution (less than 4 km), CPMs can represent convection explicitly, without using the parameterisation of deep convection, and thus reduce the model uncertainty (Fosser et al., 2024)."

- line 64, "added-value" no hyphen

Thank you, we have modified it.

- line 71, "as well as the hectares devoted to viticulture" can be concise: "planting area"

Thank you, we have rephrased it: "the hectares of vines".

- line 73, are "LOM" and "TOS" used a lot later in the text, if it is not, you may consider not use these acronyms since there are already many.

Thank you for the suggestions, however we have decided to keep the acronyms because they streamline the text and align with the cited literature (Massano et al., 2023).

Massano, L., Fosser, G., Gaetani, M., & Bois, B. (2023). Assessment of climate impact on grape productivity: A new application for bioclimatic indices in Italy. Science of the Total Environment, 905. https://doi.org/10.1016/j.scitotenv.2023.167134

- line 77, "thanks to" same as the above

Thank you, we have rephrased it.

- line 79, same as the comment in line 73

Thank you for the suggestions, please refer to the response to line 73 comment.

- line 94, "the grape yield", remove "the"

We have modified it. We thank the reviewer for spotting the error.

- line 98, , after e.g. "e.g.,"

We have modified it. We thank the reviewer for spotting the error.

- line 99, "the productivity", remove "the"

We have modified it. We thank the reviewer for spotting the error.

- line 98, remove "the" before productivity

We have modified it. We thank the reviewer for spotting the error.

- line 99, remove "the"

We have modified it. We thank the reviewer for spotting the error.

- line 100, provide data of ...

Thank you, we have modified it.

- line 100, "the area devoted to vines" please rephrase this.

Thank you, we have modified it: "vintage area".

- line 100 -104, missing "the"

Thank you, we have modified it.

- line 103, and at the national scale.

Thank you, we have modified it.

- I will stop commenting on the usage of articles.

Thanks for your comments, we have checked the manuscript for misuses of the articles.

- line 689, with? Do you mean "with"

Yes, we have corrected it.

- line 129, delete "the period"

We have done it, thank you.

- line 218, "RSME" - RMSE, also, explain what it is when you mention it the first time.

We have done it, thank you.

- line 218, "the percentage differences of RMSE with the mean of the reference" this is the normalized RMSE

Thank for pointing that out. We have modified the text and table accordingly. Furthermore, in the revised version of the paper we have opted to normalise the RMSR with respect to the range of values. The normalization method of RMSE is descripted as: "[...] Normalised Root Mean Square Error (NRMSE) with respect to the range of values, i.e. the maximum value of the variable considered ($y_{max}$) minus the minimum value ($y_{min}$), for the reference datasets (SPHERA and E-OBS)."

- line 219, remove "the". I will stop commenting on the usage of articles.

Thanks for your comments, we have checked the manuscript for misuses of the articles.

- In all line plots, please change your legends indicating the right datasets (dashed, solid)

We have done it, thank you.

- The ticks and legends are barely visible in the plots of the appendix.

Figures in the appendix have been improved.

- line 295, "for some of the temperature-based indices"

We have modified it.

- line 295, "few cases"

Thank you we have modified it.

- line 346, "in other datasets"

Thank you we have modified it.

---

## Author Comment (AC2)

**Preliminary remarks**

As suggested by Anonymous Referee #2, the tittle of the manuscript has been modified in "Using a convection-permitting climate model to assess wine grape productivity: two case studies in Italy" to better reflect the aim of the paper.

In addressing the reviewers' comments, we decided to modify the structure of the article. Different sections of the article have been moved and streamlined, to account for the suggestions of both reviewers and improve readability and the fluency of the text.

The most relevant changes are reported below:

As suggested by Reviewer #1, the Abstract has been rephrased to enhance its precision and focus.

2. Data and Methodology: Subsection "2.6 Trend Analysis" has been added, and the order of the two subsections, "2.4 Bioclimatic indices" and "2.5 Validation of climate simulations and calculation of bioclimatic indices" has been reversed. Section "2.5 Validation of climate simulations and calculation of bioclimatic indices" has been renamed "2.5 Validation of climate simulations" to better reflect its content. The revised order is now as follows: "2.4 Validation of climate simulations "; "2.5 Bioclimatic indices"; "2.6 Trend Analysis".

3. Results: Subsection "3.1 Validation of the climate simulations" has been streamlined, Table 2 was replaced by two new figures, Figure 6 and 7, to increase the readability. Figure A2 and A3 showing the time series of mean (TM), maximum Temperature (TX), minimum (TN) temperature and precipitation (P) over FRA and MON area have been moved to the main text before Figures 2 and 3showing the time series of the bioclimatic indices from 2000-2018 averaged on the FRA and MON consortium area.

At last, the "4 Discussion and Conclusion" section has undergone a comprehensive revision and expansion.

Further details are set out in our point-by-point response (in blue), which can be found below.

**Response to RC-2**

Manuscript number: egusphere-2024-941

Title: Using a convection-permitting climate model to predict wine grape productivity: two case studies in Italy

Authors: Laura T. Massano, Giorgia Fosser, Marco Gaetani, Cécile Caillaud

**Summary**

The manuscript explores the impact of convection-permitting climate model data on assessing wine grape productivity. The analysis utilizes observed wine productivity data from two consortia located in Italy. First, the climate data are compared with both meteorological observations and reanalyses to evaluate their quality. Second, single and multiple regression analyses are conducted to investigate the potential of climate-related indices to "predict" wine

productivity. The results show correlations between bioclimatic indices (in particular temperature-based indices) and wine productivity, suggesting their potential use in assessing future changes.

**General comments:**

The manuscript is well written, data and methods are described adequately; the topic treated adheres to the journal's scope. In my opinion, the manuscript would benefit from a more extensive discussion comparing the findings to previous similar studies or applications of CPM climate data in agriculture. Additionally, providing in-depth comments on the potential impact of this research, particularly its implications for future grape productivity, would underscore the relevance of the study. My general assessment is that the manuscript doesn't have any relevant flaws that prevents its publication. My recommendation is to accept the paper, provided that the specific remarks below are addressed.

English is not my native language and I have no comments on it.

We thank the Reviewer for taking the time to revise our manuscript and provide constructive comments, which helped in improving the paper. Please find below our point-by-point response, highlighted in blue.

**TITLE**

I suggest modifying the title, e.g., from "Using a convection-permitting climate model to predict wine ..." to "Using a convection-permitting climate model to assess/estimate wine grape productivity: two case studies in Italy". In fact, the authors make use of single/multiple regression to explain the variance of wine productivity data and not to predict them.

Thanks for this suggestion, we have modified the title in *"Using a convection-permitting climate model to assess wine grape productivity: two case studies in Italy"* to better fit the content of the paper.

**ABSTRACT**

**General comment**

lines 7-9: "Viticulture is tied to climate, it influences the suitability of an area, yield and quality of wine grapes. Therefore, traditional wine-growing regions could be threatened by a changing climate. Italy is at-risk being part of the Mediterranean climatic hotspot and judged in 2022 the second-largest exporter of wine worldwide."I suggest removing this sentence as the opening statement. It is redundant in the Abstract but appropriate for the Introduction section.

The sentence has been removed from the abstract and integrated into the introduction, as follow:

Introduction: "Viticulture is tied to climate, that influences the suitability of an area, the yield and quality of wine grapes.  The wine industry has a significant socio-economic influence and is a key agricultural sector in Italy. In 2022, Italy was the world's leading wine producer (49.8 million hl), and the second largest wine exporter, with a value of 7.8 billion euros (OIV, 2023)."

The Abstract now reads:

Abstract: "The article explores the potential use of climate models to reproduce wine grape productivity at local scale in Italy. To this end, both single and multiple regression approaches are used to link productivity data provided by two Italian wine consortia with bioclimatic indices. Temperature and precipitation-based bioclimatic indices are computed using the observational dataset E-OBS, the high-resolution climate reanalysis product SPHERA, the regional climate model CNRM-ALADIN and the km-scale convection-permitting climate model CNRM-AROME. The multiple regression method outperforms the single regression systematically enhancing the ability of bioclimatic indices to explain productivity variability. The results show that productivity is strongly tied with temperature-based bioclimatic indices in the area of "Consorzio per la tutela del Franciacorta" in northern Italy, while for the "Consorzio del Vino Nobile di Montepulciano" area in central Italy both temperature and precipitation-based indices are relevant. Climate models, providing similar results as E-OBS and SPHERA, appear to be a useful tool to explain productivity variance. In particular, the added value of convection-permitting resolution is evident when precipitation-based indices are considered. This assessment shows windows of opportunity for using climate models, especially at convection-permitting scale, to investigate future climate change impact on wine production."

**Specific comments**

Line 13: "and both the Regional and the Convection-permitting..." which models? If not detailing them, use the article "a" instead of "the".

The names of the analysed models are now indicated (CNRM-ALADIN and CNRM-AROME): "[...] the regional climate model CNRM-ALADIN and the km-scale convection-permitting climate model CNRM-AROME."

Line 14: "The potential of CPMs", which CPMs? One CPM or multiple CPMs? Remove the "s"

We extensively rephrase the abstract, now this concept is expressed as: "[...] the added value of convection-permitting resolution [...]". Please see the response to the comment to lines 7-9.

line 17: "of CPM, became" remove the comma "," which currently lies between the subject and the verb.

We thank you the reviewer for spotting the error.

line 17: "of CPM, became" mixed use of present and past. Please change to "becomes".

We thank you the reviewer for spotting the error.

1. INTRODUCTION

**General comment**

The Introduction is adequate and presents the state-of-the-art and the innovative approach of the research (from line 50 to the end of the section). As stated previously and for the reasons claimed by the authors in the Introduction (see lines 65-66: "Single and multiple regression approaches are used to determine the extent to which bioclimatic indices can explain changes

in wine grape productivity at the local scale"), I suggest modifying the title by removing the word 'predict' and replacing it with a more generic term such as 'assess' or 'estimate.' In fact, predicting implies providing the accuracy of the predictions, including associated errors and uncertainties, rather than just the R-squared value of the regression.

We agree with the reviewer and have modified the title as suggested.

**Specific comments**

line 21: I would start the Section with the first sentence of the Abstract, which I found redundant in that context.

The first paragraph of the introduction has been modified, and the redundance fixed:

"Viticulture is tied to climate, that influences the suitability of an area, the yield and quality of wine grapes. The wine industry has a significant socio-economic influence and is a key agricultural sector in Italy. In 2022, Italy was the world's leading wine producer (49.8 million hl), and the second largest wine exporter, with a value of 7.8 billion euros (OIV, 2023)."

Line 21-22: "Wine-growing has a strong socio-economic impact and is one of the principal agricultural economic activities in Italy, that in 2022…" I would change to "Wine-growing has … in Italy. In 2022…"

Thank you for the suggestion, we have modified as suggested.

line 22: "world's leading wine producer (49.8 million hl)" Can the authors provide any reference to support this statement?

We have added the reference: OIV, 2023: STATE OF THE WORLD VINE AND WINE SECTOR IN 2022.

line 22: "and second" change to "and the second"

Thank you for spotting the error, we have corrected.

line 27: "than the global average (Bernetti et al., 2012;.." Can you provide a more climatic-sound references to support such sentence? Further, these two references could be moved below (i.e., line 28)

Following the Reviewer's suggestion, we have added the following references:

*Giorgi, F. (2006). Climate change hot-spots. Geophysical Research Letters, 33(8), 1–4. https://doi.org/10.1029/2006GL025734*

*Roehrdanz, P. R., & Hannah, L. (2016). Climate Change, California Wine, and Wildlife Habitat. Journal of Wine Economics, 11(1), 69–87. https://doi.org/10.1017/jwe.2014.31*

*Santillán, D., Garrote, L., Iglesias, A., & Sotes, V. (2020). Climate change risks and adaptation: new indicators for Mediterranean viticulture. Mitigation and Adaptation Strategies for Global Change, 25(5), 881–899. https://doi.org/10.1007/s11027-019-09899-w*

line 31: "when frost events are still frequent" I think the word "still" can be removed to streamline the sentence

The sentence has been rephrased in response to a comment from Reviewer#1.

"Since temperature is the primary driver for the phenological phases (Fraga et al., 2016), a warmer climate may lead to a shorter growing cycle and an earlier onset of phenological phases, which would increase frost-related risk (Lamichhane, 2021; Trought et al., 1999). In fact, budburst is the most vulnerable phase to frost in the vine growing cycle, and an earlier budburst in spring would increase the exposure of the vine to late frost events.".

line 33: "are expected to experience important shifts in viticulture suitability that can consequently causes a decline in production" "causes" or "cause"? What is the subject? "shifts" or suitability?

The sentence has been rephrased:

"Furthermore, climate conditions typical of traditional wine-producing regions, such as Douro in Portugal, La Rioja in Spain, Bordeaux in France, and Tuscany in Italy, are expected to shift northwards or at higher altitude, and this modifications in viticulture suitability may consequently causes a decline in production (Adão et al., 2023; Rafique et al., 2023; Sgubin et al., 2023; Tóth and Végvári, 2016)."

Line 36: "developed" change to "computed" or something else

The word has been changed as suggested

line 48: modify ") ("

The sentence was modified to improve readability:

"To valorise the designation of origin and guarantee a defined level of quality (Gori and Alampi Sottini, 2014; Ugaglia et al., 2019), producers are organized in wine consortia (Consorzi di Tutela) according to the EU and national regulations, i.e. Regulation (EU) No 1308/2013."

line 65: "are used to determine the extent to which" change to "are used to determine to which extent"?

We have changed the sentence as suggested:

"Single and multiple regression approaches are used to determine to which extent bioclimatic indices can explain changes in wine grape productivity at local scale."

**2. Data and Methods**

**General comment**

I think it should be explained better why the authors used Aladin/Arome model outputs rather than SPHERA or other regional CP reanalyses. Can the authors stress the differences in the experimental design of numerical simulations? As stated in the manuscript, Aladin model is fed by ERA-Interim data, whereas SPHERA is fed by ERA5. Which is the difference? SPHERA are started frequently (once a day?) and receive boundary conditions every hour. What about the

Aladin/Arome numerical architecture? The authors should give the audience a taste of the differences without delving into the suggested bibliography.

The major difference between climate models (GCM, RCM or CPM) and any reanalysis dataset is that the latter assimilate observations and accurately reproduce the observed temporal variability, at least for the variables that are assimilated. Clearly the shortcoming of any reanalysis datasets is that cannot be used for climate projections. In our work, we aim to prove that also RCMs, and especially CPMs, can be used for evaluating the impact of climate on viticulture. This would open the path for studying the impact of future climate change on wine productivity.

We have clarified this aspect and we have added further details on SPHERA reanalysis dataset as well as on the climate models. In particular, Section 2.2 "Climate observations and reanalysis data" has been improved with additional information regarding the data employed and a more comprehensive justification for the selection of these specific data sources. Please find the relevant changes reported below:

[revised manuscript text omitted]

**Specific comments**

Figure 1: In both the digital and hardcopy versions of the manuscript, the geographical locations of the two consortia are difficult to discern from the images. Could the authors provide larger images and/or magnify the map of Italy?

Figure 1 has been modified to improve readability.

Line 102: "at regional level between 1994 and 2000; at national scale while" change to "at the regional level between 1994 and 2000; at the national scale while"

We have modified as suggested.

line 103: "at national scale while from 2000 to 2005" sentence not clear. Perhaps just remove "while"

We have modified as suggested.

line 112: remove the acronym "NMHSs" since it is used only once

We have removed the acronym as suggested.

line 124: "SPHERA reanalysis" change to "The SPHERA reanalysis" or just "SPHERA"

We have modified as suggested.

line 124: SPHERA is validated against a gridded dataset made of independent rain-gauges. ERA5 data are used as a benchmark. Please specify better.

Thank you for comment. We have clarified as follow:

"When validated against independent rain gauge observations, SPHERA showed an improved representation of the precipitation field, both at daily and hourly scales, compared to its driver, i.e. ERA5 (Giordani et al. 2023)."

line 138: "." is missing at the end of the text

Thank you for spotting the error. We have corrected it.

line 163: "but also take" change to "but also takes"

We have modified as suggested.

line 199: "Tests performed to investigate…are not impacted by the resolution chosen for the remapping (not shown)" I think the authors should give more details about the tests performed. At least they should say whether the tests were performed on the remapping strategy (i.e., the algorithm) or on the resolution (i.e., the final grid spacing). Please expand this point and give some details

We modified the sentence as follow:

"To compare datasets with different horizontal resolutions on equal terms (Berg et al. 2013), observations, reanalysis and model simulations are conservatively remapped on a common grid, i.e. the E-OBS regular grid at ~11 km, the coarsest among all. Tests performed to assess the impact of upscaling SPHERA and CPM at a coarser resolution showed no significant changes in the results (not shown)."

line 205: "weighing" please replace with "weighting"

Thank you for spotting the error. We have corrected it.

line 214: "SPHERA and E-OBS time series together provide a range within which the CPM and the RCM time series are expected to fall, similar to a 'confidence interval'." I disagree; can the authors support this statement by providing evidence?

We agree with the Reviewer that the sentence was confusing, and it has been removed. The whole paragraph has been rephrased:

"The comparison between climate model simulations and the reference datasets is carried out by computing the Spearman correlation, the Root Mean Square Error (RMSE) and the Normalised Root Mean Square Error (NRMSE) with respect to the range of values, i.e. the maximum value of the variable considered ($y_{max}$) minus the minimum value ($y_{min}$), for the reference datasets (SPHERA and E-OBS). In particular, the Spearman correlation coefficient is used to assess the ability of the climate models in reproducing the climate variability of the reference datasets, while RMSE and NRMSE provides a measure of the climate models biases. Moreover, the statistical significance of the model biases is assessed by applying a Welch's two-tailed t-test (Welch, 1938), with a 95% level of confidence."

Line 217: I don't get why E-OBS are used within the parenthesis Line 221: too many "()"

We have corrected it as suggested.

**3. Results**

**General comment**

I don't see why many Figures/Tables that are commented in this Section are taken from the Appendix (e.g., lines 255, 258, 261, 265, 267, 277, 298, 306, 316). This doesn't help the readability of the manuscript. I encourage the authors to rethink this section. For example, it should start from line 269 "Figure 2 and Figure 3 show the ten bioclimatic..." and the first sentence "The precipitation and temperature time series of both..." could be moved elsewhere in the manuscript (Discussion or Appendix). Alternatively, some tables/plots shown in the Appendix could be streamlined (e.g., Table A 4 which has many columns and rows) or simply removed and their content moved into the text (e.g., Table A 6, Table A 8). Table 2 is hard/difficult to read. I wonder whether a plot could help its readability. I suggest the authors to reconsider it. If they decide to keep it, I suggest to remove the RMSE column, since it is not commented in the text. Further, it is shown the RMSE% which is more informative since the ranges of bioclimatic indexes are very different.

Following the Reviewer' suggestion, we have rethanked this section.

The first paragraph of this section has been streamlined and now reads:

"Prior to the computation of the bioclimatic indices, the precipitation and temperature fields in both consortia (FRA and MON) are analysed to assess the potential biases, which could impact on the temperature and precipitation-based bioclimatic indices. Figure 2 for FRA and Figure 3 for MON show the precipitation (P) and temperature (TM: mean temperature, TX: max temperature and TN: min temperature) time series of E-OBS, SPHERA, RCM and CPM for the period 2000-2018. In general, both RCM and CPM well reproduce SPHERA temporal variability as also confirmed by the high and significant correlations for all the climate variables in both consortia (Table A 2). Nevertheless, both climate models tend to overestimate mean and maximum temperature while underestimating minimum temperature, as reflected by the statistical differences in mean values (Table A 3). Both climate models, and especially the RCM, underestimate precipitation in FRA, while the CPM tends to overestimate it in MON. In FRA the variability observed in E-OBS is always reproduced in both climate simulations with RCM being closer to E-OBS mean values than CPM (Table A3). However, in MON, E-OBS minimum temperature time series shows a strong decrease of almost 2°C between 2015 and 2018 (Figure 3), which is not observed in any models nor SPHERA. Further investigations revealed that this temperature decline is observed throughout the entire TOS and is inconsistent with other observational records (not shown). This E-OBS misrepresentation of the temperature field has a subsequent effect on the mean temperature time series (Figure 3), the temporal correlations (Table A 2), and is likely to be reflected in the temperature-based bioclimatic indices in TOS region, and at local scale in MON."

Table 2 has been replaced by Figures 4 and 5, which show the correlation coefficient and the NRMSE for each index of the comparison between the climatic sources used, for FRA and MON respectively.

Moreover, Table A6 and A8 have been aggregated in Table A7 that now contains the Sen's slope of productivity in both FRA and MON consortia. "Table A7: Sen's slope of the productivity in FRA and MON. Sen's slope is a statistical measure used to calculate the rate of change in a variable over time, based on the Sen's estimator. Asterisk (*) indicate a significant trend (p<=0.05)"

**Specific comments**

line 276: "(Table 2)" remove empty space

Thank you, we have removed them.

line 296: "CNI with model simulations" change to "CNI with model climate simulations" or simply "CNI with climate simulations", in fact, SPHERA is a model simulation too.

Changed as suggested: "climate model simulations".

lines 302-306: I would like to see this paragraph in the Discussion section, where it is more pertinent.

We have removed the paragraphs 302-306 and 325-330 from this section as suggested. The concept that was presented here is now addressed in the Discussion section; the relevant paragraph is reported below:

"These results, which are obtained at the local scale using data from wine consortia, complement and expand the previous study conducted at the regional scale by Massano et al. (2023) using ISTAT productivity data and E-OBS (v26, resolution ~11 km) climate data. In fact, they did not find any statistically significant correlations for LOM or TOS region, where FRA and MON respectively lie, for neither with temperature-based nor precipitation-based indices. At the contrary, in this work the MR can explain up to 64% in FRA with RCM and 45% in MON with the CPM. This indicates that working at a local scale and including a larger variety of bioclimatic indices is crucial to improve the portion of productivity variance explained by the bioclimatic indices."

lines 325-330: as above

We have modified the paragraph following the Reviewer's comment. Please refer to the response to the comment on lines 302-306 for further details.

Table 3: Can the authors discuss why, in MON, the variance of E-OBS is 44% for the SR case and 32% for MR? Is it related to the poor quality of E-OBS data (in MON) as argued previously? This happens also for RCM although to a smaller extent (32%→29%). I would like to see a plausible explanation in the Discussion section

The variance in SR is simply calculated as the square of the Spearman correlation between the single bioclimatic index and the observed productivity. The MR uses the k-fold method to select the relevant bioclimatic indices and determine the coefficients to predict the productivity with a formula such as productivity = a1*index1 + a2*index2 + a3*index3 + ... We then calculate the variance as the square of the Pearson correlation between the predicted and the observed productivity.

In case only one bioclimatic index is selected, the correlation (and therefore the variance) might be different from the SR method. In fact, with the MR the correlation is calculated between observed productivity and $\alpha$*Index1 (i.e. the predicted productivity) and not between observed productivity and Index1 as in SR. in addition, the SR and MR method use a slightly different type of correlation, Spearman in the former and Pearson in the latter.

We have added a paragraph in the Data and Methodology section to better explain it, and we have modified the caption of Table 2 to recall the definition.

"The so-optimised MR model (productivity = a1\*index1 + a2\*index2 + a3\*index3 + ..., with indexn indicating the selected bioclimatic index) is then used to predict the productivity and the Pearson correlation between predicted and observed productivity is calculated. Following Massano et al. (2023), the comparison between the SR and MR methods is performed in terms of the productivity variance explained by the prediction, estimated by computing the coefficient of determination, i.e. the square of the correlation coefficient."

We have also clarified in the discussion between SR and MR performance in section 3.2.2: "To note that the decrease in performance from SR to MR method only occurs when only one bio-climatic index is selected in the MR. This could be linked to coefficient included in the MR (i.e. productivity=a1\*Index1) or to the different type of correlation used in SR (Spearman) and MR (Pearson)."

And again, in the Discussion and conclusion

"When more than one bioclimatic index is relevant, the multi-regression method outperforms the single regression approach, systematically enhancing the explanatory power of bioclimatic indices regarding productivity variability. Furthermore, the method has the potential to deliver predictors that are fit for purpose."

**4. Discussion and conclusion**

**General comment**

I think this section lacks a critical review of the results found in comparison with previous studies. It looks like a summary of the manuscript. If no or few previous studies are found, it should stress the novelty of the study and highlighted the potential and limits of CP model data in assessing productivity.

Further, the main advantage of using CPM data is the removal of any parameterisation to model convection processes. Indeed, it is well known they provide more accurate precipitation estimates than RCM data (e.g., lines 53-55). However, you found that wine productivity is mostly related to temperature-based bioclimatic indexes rather than precipitation-based one. Do the authors have any comment on it?

We are grateful to the Reviewer for drawing attention to the shortcomings of this section, which drove a comprehensive revision and expansion of the part.

In particular, in order to stress the novelty of the work here presented, we have added the following paragraph in the Introduction section:

"Nevertheless, no prior studies have employed CPMs to examine the influence of climate variability and change on viticulture.

The present study presents a novel approach to estimate wine grape productivity at the local scale by using a CPM, showing windows of opportunity for the use of CPMs in the context of ongoing and future climate change."

And in the Discussion and conclusion section:

"This study represents, to the best of the authors' knowledge, the first application of a CPM to investigate the impact of climate variability and change on wine grape productivity, through the use of bioclimatic indices."

The predominance of the impact of temperature over precipitation on productivity is a fact in viticulture. We have addressed it in the introduction:

"Since temperature is the primary driver for the phenological phases (Fraga et al., 2016), a warmer climate may lead to a shorter growing cycle and an earlier onset of phenological phases, which would increase frost-related risk (Lamichhane, 2021; Trought et al., 1999). In fact, budburst is the most vulnerable phase to frost in the vine growing cycle, and an earlier budburst in spring would increase the exposure of the vine to late frost events."

Furthermore, this aspect is considered in in the Discussion and conclusion section, when the limitations of the CPM in this particular application are discussed:

"When the MR approach is applied, climate models appear to be a useful tool to explain the variability of productivity, improving the results obtained using E-OBS. However, the use of the CPM does not show a clear added value with respect to the RCM, since it performs better in MON, but not in FRA. This could be linked to the fact that temperature is generally the main driver of wine grape production, and the added value of the CPM become more evident when precipitation is a dominant factor, as in MON. Nevertheless, in a changing climate, with precipitation frequency and intensity expected to change (Tramblay and Somot, 2018; Zittis et al., 2021), the relevance of precipitation, along with precipitation-based bioclimatic indices, on grape productivity might increase and in turn the use of CPM might become crucial."

**Specific comments**

line 401: "This could be link" replace with "This could be linked"

Thank you for spotting the error, we have corrected it.

References

line 573: incomplete reference

Thank you for spotting the error, we have corrected it.

**Appendix A**

Figure A2 provided in poor quality when on hardcopy

The quality of the figure has been improved and it has been moved to the main text in Section 3.1 "Validation of the climate simulations".

---

## Author Response (AR3)

**Dear Reviewer,**

Thank you for your feedback on our manuscript. We appreciate the time and effort you invested in reviewing our work. We have considered your suggestions and have implemented the minor changes you recommended. Further details are set out in our point-by-point response (in blue), which can be found below.

**Response to RC-1**

Line 249: It is good that you italicized this approach.

**Thank you, the reviewer's feedback is much appreciated.**

Table 3: Thank you for the detailed explanation. From your comments, I understand that in the specific case of FRA, GSP is important, but not as important as other factors, such as CNI. GSP may appear significant in one dataset but not in another. CPM does not necessarily generate 100% precipitation-related results and can also be spatially distributed. Although I acknowledge this possibility, I would recommend that the authors suggest future work to test with more datasets.

Table 3, line 384-385:

After:

"Thus, for MON, the improved representation of the precipitation field at convection-permitting scale could be a relevant factor, since in other datasets at coarser resolution (i.e. E-OBS and RCM) precipitation-based indices are excluded by the MR."

We added:

"To improve the understanding of this aspect and clarify the relative importance of the precipitationbased indices for the two study areas, the same methodology employed here could be applied to other climatic datasets derived from different convection-permitting models."